# Natural Compression for Distributed Deep Learning

## Abstract

Modern deep learning models are often trained in parallel over a collection of distributed machines to reduce training time. In such settings, communication of model updates among machines becomes a significant performance bottleneck and various lossy update compression techniques have been proposed to alleviate this problem. In this work, we introduce a new, simple yet theoretically and practically effective compression technique: *natural compression ($\mathcal{C}_{\mathrm{nat}}$)*. Our technique is applied individually to all entries of the to-be-compressed update vector and works by randomized rounding to the nearest (negative or positive) power of two, which can be computed in a "natural" way by ignoring the mantissa. We show that compared to no compression, $\mathcal{C}_{\mathrm{nat}}$ increases the second moment of the compressed vector by not more than the tiny factor $9/8$, which means that the effect of $\mathcal{C}_{\mathrm{nat}}$ on the convergence speed of popular training algorithms, such as distributed SGD, is negligible. However, the communications savings enabled by $\mathcal{C}_{\mathrm{nat}}$ are substantial, leading to $3$-$4\times$ *improvement in overall theoretical running time*. For applications requiring more aggressive compression, we generalize $\mathcal{C}_{\mathrm{nat}}$ to *natural dithering*, which we prove is *exponentially better* than the common random dithering technique. Our compression operators can be used on their own or in combination with existing operators for a more aggressive combined effect, and offer new state-of-the-art both in theory and practice.

## 1 Introduction

Modern deep learning models (He et al., 2016) are almost invariably trained in parallel or distributed environments, which is necessitated by the enormous size of the data sets and dimension and complexity of the models required to obtain state-of-the-art performance. In our work, the focus is on the *data-parallel* paradigm, in which the training data is split across several workers capable of operating in parallel (Bekkerman et al., 2011; Recht et al., 2011). Formally, we consider optimization problems of the form

$$\min_{x \in \mathbb{R}^d} f(x) := \frac{1}{n} \sum_{i=1}^{n} f_i(x), \tag{1}$$

where $x \in \mathbb{R}^d$ represents the parameters of the model, $n$ is the number of workers, and $f_i : \mathbb{R}^d \to \mathbb{R}$ is a loss function composed of data stored on worker $i$. Typically, $f_i$ is modeled as a function of the form $f_i(x) := \mathrm{E}_{\zeta \sim \mathcal{D}_i} [f_\zeta(x)]$, where $\mathcal{D}_i$ is the distribution of data stored on worker $i$, and $f_\zeta : \mathbb{R}^d \to \mathbb{R}$ is the loss of model $x$ on data point $\zeta$. The distributions $\mathcal{D}_1, \ldots, \mathcal{D}_n$ can be different on every node, which means that the functions $f_1, \ldots, f_n$ may have different minimizers. This framework covers i) stochastic optimization when either $n = 1$ or all $\mathcal{D}_i$ are identical, and ii) empirical risk minimization when $f_i(x)$ can be expressed as a finite average, i.e, $\frac{1}{m_i} \sum_{i=1}^{m_i} f_{ij}(x)$ for some $f_{ij} : \mathbb{R}^d \to \mathbb{R}$.

**Distributed Learning.** Typically, problem (1) is solved by distributed stochastic gradient descent (SGD) (Robbins & Monro, 1951), which works as follows: Stochastic gradients $g_i(x^k)$'s are computed locally and sent to a master node, which performs update aggregation $g^k = \sum_i g_i(x^k)$. The aggregated gradient $g^k$ is sent back to the workers and each performs a single step of SGD: $x^{k+1} = x^k - \frac{\eta^k}{n} g^k$, where $\eta^k > 0$ is a step size.

A key bottleneck of the above algorithm, and of its many variants (e.g., variants utilizing mini-batching (Goyal et al., 2017), importance sampling (Horváth & Richtárik, 2019), momentum (Nesterov, 2013), or variance reduction (Johnson & Zhang, 2013)), is the cost of communication of the

typically dense gradient vector $g_i(x^k)$, and in a parameter-sever implementation with a master node, also the cost of broadcasting the aggregated gradient $g^k$. These are $d$ dimensional vectors of floats, with $d$ being very large in modern deep learning. It is well-known (Seide et al., 2014; Alistarh et al., 2017; Zhang et al., 2017; Lin et al., 2018; Lim et al., 2018) that in many practical applications with common computing architectures, communication takes much more time than computation, creating a bottleneck of the entire training system.

**Communication Reduction.** Several solutions were suggested in the literature as a remedy to this problem. In one strain of work, the issue is addressed by giving each worker "more work" to do, which results in a better communication-to-computation ratio. For example, one may use mini-batching to construct more powerful gradient estimators (Goyal et al., 2017), define local problems for each worker to be solved by a more advanced local solver (Shamir et al., 2014; Richtárik & Takáč, 2016; Reddi et al., 2016), or reduce communication frequency (e.g., by communicating only once (McDonald et al., 2009; Zinkevich et al., 2010) or once every few iterations (Stich, 2018)). An orthogonal approach to the above efforts aims to reduce the size of the communicated vectors instead (Seide et al., 2014; Alistarh et al., 2017; Wen et al., 2017; Wangni et al., 2018; Hubara et al., 2017) using various lossy (and often randomized) *compression* mechanisms, commonly known in the literature as quantization techniques. In their most basic form, these schemes decrease the # bits used to represent floating point numbers forming the communicated $d$-dimensional vectors (Gupta et al., 2015; Na et al., 2017), thus reducing the size of the communicated message by a constant factor. Another possibility is to apply randomized *sparsification* masks to the gradients (Suresh et al., 2017; Konečný & Richtárik, 2018; Alistarh et al., 2018; Stich et al., 2018), or to rely on coordinate/block descent updates-rules, which are sparse by design (Fercoq et al., 2014).

One of the most important considerations in the area of compression operators is the *compression-variance* trade-off (Konečný & Richtárik, 2018; Alistarh et al., 2017; Horváth et al., 2019). For instance, while random dithering approaches attain up to $\mathcal{O}(d^{1/2})$ compression (Seide et al., 2014; Alistarh et al., 2017; Wen et al., 2017), the most aggressive schemes reach $\mathcal{O}(d)$ compression by sending a constant number of bits per iteration only (Suresh et al., 2017; Konečný & Richtárik, 2018; Alistarh et al., 2018; Stich et al., 2018). However, the more compression is applied, the more information is lost, and the more will the quantized vector differ from the original vector we want to communicate, increasing its statistical variance. Higher variance implies slower convergence (Alistarh et al., 2017; Mishchenko et al., 2019), i.e., more communication rounds. So, ultimately, compression approaches offer a trade-off between the communication cost per iteration and the number of communication rounds.

Outside of the optimization for machine learning, compression operators are very relevant to optimal quantization theory and control theory (Elia & Mitter, 2001; Sun & Goyal, 2011; Sun et al., 2012).

**Summary of Contributions.** The key contributions of this work are following:

• **New compression operators.** We construct a new *"natural compression"* operator ($\mathcal{C}_{\mathrm{nat}}$; see Sec. 2) based on a randomized rounding scheme in which each float of the compressed vector is rounded to a (positive or negative) power of 2. This compression has a provably small variance, at most $1/8$ (see Thm 1), which implies that theoretical convergence results of SGD-type methods are essentially unaffected (see Thm 6). At the same time, substantial savings are obtained in the amount of communicated bits per iteration ($3.56\times$ less for *float32* and $5.82\times$ less for *float64*). In addition, we utilize these insights and develop a new random dithering operator—*natural dithering* ($\mathcal{D}_{\mathrm{nat}}^{p,s}$; see Sec. 3)—which is *exponentially better* than the very popular "standard" random dithering operator (see Thm 5). We remark that $\mathcal{C}_{\mathrm{nat}}$ and the identity operator arise as limits of $\mathcal{D}_{\mathrm{nat}}^{p,s}$ and $\mathcal{D}_{\mathrm{sta}}^{p,s}$ as $s \to \infty$, respectively. Importantly, our new compression techniques can be *combined* with existing compression and sparsification operators for a more dramatic effect as we argued before.

• **State-of-the-art compression.** When compared to previous state-of-the-art compressors such as (any variant of) sparsification and dithering—techniques used in methods such as Deep Gradient Compression (Lin et al., 2018), QSGD (Alistarh et al., 2017) and TernGrad (Wen et al., 2017)—our compression operators offer provable and often large improvements in practice, thus leading to *new state of the art*. In particular, given a budget on the second moment $\omega + 1$ (see Eq (3)) of a compression operator, which is the main factor influencing the increase in the number of communications when communication compression is applied compared to no compression, our compression operators offer the largest compression factor, resulting in fewest bits transmitted (see Fig 1).

• **Lightweight & simple low-level implementation.** We show that apart from a randomization procedure (which is inherent in all unbiased compression operators), natural compression is *computation-free*. Indeed, natural compression essentially amounts to the trimming of the mantissa and possibly

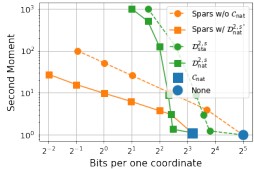

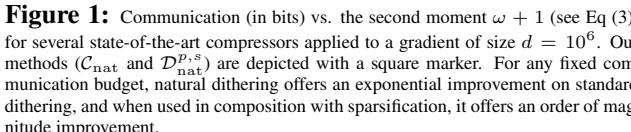

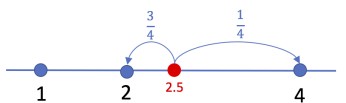

**Figure 1:** Communication (in bits) vs. the second moment $\omega + 1$ (see Eq (3)) for several state-of-the-art compressors applied to a gradient of size $d = 10^6$. Our methods ($\mathcal{C}_{\mathrm{nat}}$ and $\mathcal{D}_{\mathrm{nat}}^{p,s}$) are depicted with a square marker. For any fixed communication budget, natural dithering offers an exponential improvement on standard dithering, and when used in composition with sparsification, it offers an order of magnitude improvement.

**Figure 2:** An illustration of nat. compression applied to $t = 2.5$: $\mathcal{C}_{\mathrm{nat}}(2.5) = 2$ with probability $\frac{4-2.5}{2} = 0.75$, and $\mathcal{C}_{\mathrm{nat}}(2.5) = 4$ with prob. $\frac{2.5-2}{2} = 0.25$. This choice of probabilities ensures that the compression operator is unbiased, i.e., $\mathrm{E}\left[\mathcal{C}_{\mathrm{nat}}(t)\right] \equiv t$.

increasing the exponent by one. This is the first compression mechanism with such a "natural" compatibility with binary floating point types.

• **Proof-of-concept system with in-network aggregation (INA).** The recently proposed SwitchML (Sapio et al., 2019) system alleviates the communication bottleneck via in-network aggregation (INA) of gradients. Since current programmable network switches are only capable of adding integers, new update compression methods are needed which can supply outputs in an integer format. Our *natural compression* mechanism is the first that is provably able to operate in the SwitchML framework as it communicates integers only: the sign, plus the bits forming the exponent of a float. Moreover, having bounded (and small) variance, it is compatible with existing distributed training methods.

• **Bidirectional compression for SGD.** We provide convergence theory for distributed SGD which allows for *compression both at the worker and master side* (see Algorithm 1). The compression operators compatible with our theory form a large family (operators $\mathcal{C} \in \mathbb{B}(\omega)$ for some finite $\omega \geq 0$; see Definition 2). This enables safe experimentation with existing and facilitates the development of new compression operators fine-tuned to specific deep learning model architectures. Our convergence result (Thm 1) applies to smooth and non-convex functions, and our rates predict linear speed-up with respect to the number of machines.

• **Better total complexity.** Most importantly, we are the first to *prove* that the increase in the number of iterations caused by (a carefully designed) compression is more than compensated by the savings in communication, which leads to an overall provable speedup in training time. Read Thm 6, the discussion following the theorem and Table 1 for more details. To the best of our knowledge, standard dithering (QSGD (Alistarh et al., 2017)) is the only previously known compression technique able to achieve this with our distributed SGD with bi-directional compression. Importantly, our natural dithering is exponentially better than standard dithering, and hence provides for state-of-the -art performance in connection with Algorithm 1.

• **Experiments.** We show that $\mathcal{C}_{\mathrm{nat}}$ significantly reduces the training time compared to no compression. We provide empirical evidence in the form scaling experiments, showing that $\mathcal{C}_{\mathrm{nat}}$ does not hurt convergence when the number of workers is growing. We also show that popular compression methods such as random sparsification and random dithering are enhanced by combination with natural compression or natural dithering (see Appendix A). The combined compression technique reduces the number of communication rounds without any noticeable impact on convergence providing the same quality solution.

## 2 NATURAL COMPRESSION

We define a new (randomized) compression technique, which we call *natural compression*. This is fundamentally a function mapping $t \in \mathbb{R}$ to a random variable $\mathcal{C}_{\mathrm{nat}}(t) \in \mathbb{R}$. In case of vectors $x = (x_1, \ldots, x_d) \in \mathbb{R}^d$ we apply it in an element-wise fashion: $(\mathcal{C}_{\mathrm{nat}}(x))_i = \mathcal{C}_{\mathrm{nat}}(x_i)$. Natural compression $\mathcal{C}_{\mathrm{nat}}$ performs a randomized logarithmic rounding of its input $t \in \mathbb{R}$. Given nonzero $t$, let $\alpha \in \mathbb{R}$ be such that $|t| = 2^\alpha$ (i.e., $\alpha = \log_2 |t|$). Then $2^{\lfloor \alpha \rfloor} \leq |t| = 2^\alpha \leq 2^{\lceil \alpha \rceil}$ and we round $t$ to either $\mathrm{sign}(t)2^{\lfloor \alpha \rfloor}$, or to $\mathrm{sign}(t)2^{\lceil \alpha \rceil}$. When $t = 0$, we set $\mathcal{C}_{\mathrm{nat}}(0) = 0$. The probabilities are chosen so that $\mathcal{C}_{\mathrm{nat}}(t)$ is an unbiased estimator of $t$, i.e., $\mathrm{E}\left[\mathcal{C}_{\mathrm{nat}}(t)\right] = t$ for all $t$. For instance, $t = -2.75$ will be rounded to either $-4$ or $-2$ (since $-2^2 \leq -2.75 \leq -2^1$), and $t = 0.75$ will be rounded to either $^1/_2$ or $1$ (since $2^{-1} \leq 0.75 \leq 2^0$). As a consequence, if $t$ is an integer power of 2, then $\mathcal{C}_{\mathrm{nat}}$ will leave $t$ unchanged, see Fig. 2.

**Definition 1** (Natural compression). Natural compression is a random function $\mathcal{C}_{\text{nat}} : \mathbb{R} \mapsto \mathbb{R}$ defined as follows. We set $\mathcal{C}_{\text{nat}}(0) = 0$. If $t \neq 0$, we let

$$\mathcal{C}_{\text{nat}}(t) := \begin{cases} \text{sign}(t) \cdot 2^{\lfloor \log_2 |t| \rfloor}, & \text{with } p(t), \\ \text{sign}(t) \cdot 2^{\lceil \log_2 |t| \rceil}, & \text{with } 1 - p(t), \end{cases} \tag{2}$$

where probability $p(t) := \frac{2^{\lceil \log_2 |t| \rceil} - |t|}{2^{\lfloor \log_2 |t| \rfloor}}$.

Alternatively, (2) can be written as $\mathcal{C}_{\text{nat}}(t) = \text{sign}(t) \cdot 2^{\lfloor \log_2 |t| \rfloor}(1 + \lambda(t))$, where $\lambda(t) \sim$ Bernoulli$(1 - p(t))$; that is, $\lambda(t) = 1$ with prob. $1 - p(t)$ and $\lambda(t) = 0$ with prob. $p(t)$. The key properties of any (unbiased) compression operator are variance, ease of implementation, and compression level. We characterize the remarkably low variance of $\mathcal{C}_{\text{nat}}$ and describe an (almost) effortless and *natural* implementation, and the compression it offers in rest of this section.

$\mathcal{C}_{\text{nat}}$ **has a negligible variance:** $\omega = 1/8$. We identify natural compression as belonging to a large class of unbiased compression operators with bounded second moment (Jiang & Agrawal, 2018; Khirirat et al., 2018; Horváth et al., 2019), defined below.

**Definition 2** (Compression operators). A function $\mathcal{C} \colon \mathbb{R}^d \to \mathbb{R}^d$ mapping a deterministic input to a random vector is called a *compression operator* (on $\mathbb{R}^d$). We say that $\mathcal{C}$ is *unbiased* and has *bounded second moment* ($\omega \geq 0$) if

$$\mathrm{E}\left[\mathcal{C}(x)\right] = x, \quad \mathrm{E}\left\|\mathcal{C}(x)\right\|^2 \leq (\omega + 1)\left\|x\right\|^2 \qquad \forall x \in \mathbb{R}^d. \tag{3}$$

If $\mathcal{C}$ satisfies (3), we will write $\mathcal{C} \in \mathbb{B}(\omega)$.

Note that $\omega = 0$ implies $\mathcal{C}(x) = x$ almost surely. It is easy to see that the *variance* of $\mathcal{C}(x) \in \mathbb{B}(\omega)$ is bounded as: $\mathrm{E}\left\|\mathcal{C}(x) - x\right\|^2 \leq \omega \left\|x\right\|^2$. If this holds, we say that "$\mathcal{C}$ has variance $\omega$". The importance of $\mathbb{B}(\omega)$ stems from two observations. First, operators from this class are known to be compatible with several optimization algorithms (Khirirat et al., 2018; Horváth et al., 2019). Second, this class includes most compression operators used in practice (Alistarh et al., 2017; Wen et al., 2017; Wangni et al., 2018; Mishchenko et al., 2019). In general, the larger $\omega$ is, the higher compression level might be achievable, and the worse impact compression has on the convergence speed.

The main result of this section says that the natural compression operator $\mathcal{C}_{\text{nat}}$ has variance $1/8$.

**Theorem 1.** $\mathcal{C}_{\text{nat}} \in \mathbb{B}(1/8)$.

Consider now a similar unbiased randomized rounding operator to $\mathcal{C}_{\text{nat}}$; but one that rounds to one of the nearest integers (as opposed to integer powers of 2). We call it $\mathcal{C}_{\text{int}}$. At first sight, this may seem like a reasonable alternative to $\mathcal{C}_{\text{nat}}$. However, as we show next, $\mathcal{C}_{\text{int}}$ does not have a finite second moment and is hence incompatible with existing optimization methods.

**Theorem 2.** *There is no $\omega \geq 0$ such that $\mathcal{C}_{\text{int}} \in \mathbb{B}(\omega)$.*

**From 32 to 9 bits, with lightning speed.** We now explain that performing natural compression of a real number in a binary floating point format is computationally cheap. In particular, excluding the randomization step, $\mathcal{C}_{\text{nat}}$ amounts to simply dispensing off the mantissa in the binary representation. The most common computer format for real numbers, *binary*32 (resp. *binary*64) of the IEEE 754 standard, represents each number with 32 (resp. 64) bits, where the first bit represents the sign, 8 (resp. 11) bits are used for the exponent, and the remaining 23 (resp. 52) bits are used for the mantissa. A scalar $t \in \mathbb{R}$ is represented in the form $(s, e_7, e_6, \ldots, e_0, m_1, m_2, \ldots, m_{23})$, where $s, e_i, m_j \in \{0, 1\}$ are bits, via the relationship $t = (-1)^s \times 2^{e-127} \times (1 + m)$, $e = \sum_{i=0}^{7} e_i 2^i$, $m = \sum_{j=1}^{23} m_j 2^{-j}$, where $s$ is the *sign*, $e$ is the *exponent* and $m$ is the *mantissa*. A *binary*32 representation of $t = -2.75$ is visualized in Fig 4. In this case, $s = 1$, $e_7 = 1$, $m_2 = m_3 = 1$ and hence $t = (-1)^s \times 2^{e-127} \times (1 + m) = -1 \times 2 \times (1 + 2^{-2} + 2^{-3}) = -2.75$.

It is clear that $0 \leq m < 1$, and hence $2^{e-127} \leq |t| < 2^{e-126}$. Moreover, $p(t) = \frac{2^{e-126} - |t|}{2^{e-127}} = 2 - |t|2^{127-e} = 1 - m$. Hence, natural compression of $t$ represented as *binary*32 is given as follows:

$$\mathcal{C}_{\text{nat}}(t) = \begin{cases} (-1)^s \times 2^{e-127}, & \text{with probability } 1 - m, \\ (-1)^s \times 2^{e-126}, & \text{with probability } m. \end{cases}$$

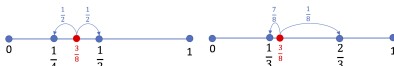

**Figure 3:** Randomized rounding for natural (left) and standard (right) dithering ($s = 3$ levels).

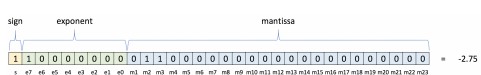

**Figure 4:** IEEE 754 single-precision binary floating-point format: *binary*32.

Observe that $(-1)^s \times 2^{e-127}$ is obtained from $t$ by setting the mantissa $m$ to zero, and keeping both the sign $s$ and exponent $e$ unchanged. Similarly, $(-1)^s \times 2^{e-126}$ is obtained from $t$ by setting the mantissa $m$ to zero, keeping the sign $s$, and increasing the exponent by one. Hence, *both values can be computed from $t$ essentially without any computation.*

**Communication savings.** In summary, in case of binary32, the output $\mathcal{C}_{\mathrm{nat}}(t)$ of natural compression is encoded using the 8 bits in the exponent and an extra bit for the sign. *This is $3.56\times$ less communication.* In case of binary64, we only need 11 bits for the exponent and 1 bit for the sign, and *this is $5.82\times$ less communication.*

**Compatibility with other compression techniques** We start with a simple but useful observation about composition of compression operators.

**Theorem 3.** *If $\mathcal{C}_1 \in \mathbb{B}(\omega_1)$ and $\mathcal{C}_2 \in \mathbb{B}(\omega_2)$, then $\mathcal{C}_1 \circ \mathcal{C}_2 \in \mathbb{B}(\omega_{12})$, where $\omega_{12} = \omega_1\omega_2 + \omega_1 + \omega_2$, and $\mathcal{C}_1 \circ \mathcal{C}_2$ is the composition defined by $(\mathcal{C}_1 \circ \mathcal{C}_2)(x) = \mathcal{C}_1(\mathcal{C}_2(x))$.*

Combining this result with Thm. 1, we observe that for any $\mathcal{C} \in \mathbb{B}(\omega)$, we have $\mathcal{C}_{\mathrm{nat}} \circ \mathcal{C} \in \mathbb{B}(9\omega/8 + 1/8)$. Since $\mathcal{C}_{\mathrm{nat}}$ offers substantial communication savings with only a negligible effect on the variance of $\mathcal{C}$, a key use for natural compression beyond applying it as the sole compression strategy is to deploy it with other effective techniques as a final compression mechanism (e.g., with sparsifiers (Stich et al., 2018)), boosting the performance of the system even further. However, our technique will be useful also as a post-compression mechanism for compressions that do not belong to $\mathbb{B}(\omega)$ (e.g., TopK sparsifier (Alistarh et al., 2018)). The same comments apply to the *natural dithering* operator $\mathcal{D}_{\mathrm{nat}}^{p,s}$, defined in the next section.

## 3  NATURAL DITHERING

Motivated by the natural compression introduced in Sec 2, here we propose a new random dithering operator which we call *natural dithering*. However, it will be useful to introduce a more general dithering operator, one generalizing both the natural and the standard dithering operators. For $1 \leq p \leq +\infty$, let $\|x\|_p$ be $p$-norm: $\|x\|_p \coloneqq \left(\sum_i |x_i|^p\right)^{1/p}$.

**Definition 3** (General dithering). The *general dithering* operator with respect to the $p$ norm and with $s$ levels $0 = l_s < l_{s-1} < l_{s-2} < \cdots < l_1 < l_0 = 1$, denoted $\mathcal{D}_{\mathrm{gen}}^{\mathcal{C},p,s}$, is defined as follows. Let $x \in \mathbb{R}^d$. If $x = 0$, we let $\mathcal{D}_{\mathrm{gen}}^{\mathcal{C},p,s}(x) = 0$. If $x \neq 0$, we let $y_i \coloneqq |x_i|/\|x\|_p$ for all $i \in [d]$. Assuming $l_{u+1} \leq y_i \leq l_u$ for some $u \in \{0, 1, \ldots, s-1\}$, we let $\left(\mathcal{D}_{\mathrm{gen}}^{\mathcal{C},p,s}(x)\right)_i = \mathcal{C}(\|x\|_p) \times \mathrm{sign}(x_i) \times \xi(y_i)$, where $\mathcal{C} \in \mathbb{B}(\omega)$ for some $\omega \geq 0$ and $\xi(y_i)$ is a random variable equal to $l_u$ with probability $\frac{y_i - l_{u+1}}{l_u - l_{u+1}}$, and to $l_{u+1}$ with probability $\frac{l_u - y_i}{l_u - l_{u+1}}$. Note that $\mathrm{E}\left[\xi(y_i)\right] = y_i$.

Standard (random) dithering, $\mathcal{D}_{\mathrm{sta}}^{p,s}$, (Goodall, 1951; Roberts, 1962) is obtained as a special case of general dithering (which is also novel) for a linear partition of the unit interval, $l_{s-1} = 1/s$, $l_{s-2} = 2/s$, $\ldots$, $l_1 = (s-1)/s$ and $\mathcal{C}$ equal to the identity operator. $\mathcal{D}_{\mathrm{sta}}^{2,s}$ operator was used in QSGD (Alistarh et al., 2017) and $\mathcal{D}_{\mathrm{sta}}^{\infty,1}$ in Terngrad (Wen et al., 2017). *Natural dithering*—a novel compression operator introduced in this paper—arises as a special case of general dithering for $\mathcal{C}$ being an identity operator and a binary geometric partition of the unit interval: $l_{s-1} = 2^{1-s}$, $l_{s-2} = 2^{2-s}$, $\ldots$, $l_1 = 2^{-1}$. For the INA application, we apply $\mathcal{C} = \mathcal{C}_{\mathrm{nat}}$ to have output always in powers of 2, which would introduce extra factor of $9/8$ in the second moment. A comparison of the $\xi$ operators for the standard and natural dithering with $s = 3$ levels applied to $t = 3/8$ can be found in Fig 3. When $\mathcal{D}_{\mathrm{gen}}^{\mathcal{C},p,s}$ is used to compress gradients, each worker communicates the norm (1 float), vector of signs ($d$ bits) and efficient encoding of the effective levels for each entry $i = 1, 2, \ldots, d$. Note that $\mathcal{D}_{\mathrm{nat}}^{p,s}$ is essentially an application of $\mathcal{C}_{\mathrm{nat}}$ to all normalized entries of $x$, with two differences: i) we can also communicate the compressed norm $\|x\|_p$, ii) in $\mathcal{C}_{\mathrm{nat}}$ the interval $[0, 2^{1-s}]$ is subdivided further, to machine precision, and in this

| Approach | $\mathcal{C}_{W_i}$ | No. iterations $T'(\omega_W) = \mathcal{O}((\omega_W + 1)^\theta)$ | Bits per 1 Iter. $W_i \mapsto M$ | Speedup Factor |
|---|---|---|---|---|
| Baseline | identity | 1 | $32d$ | 1 |
| **New** | $\mathcal{C}_{\mathrm{nat}}$ | $(9/8)^\theta$ | $9d$ | $3.2\times\text{–}3.6\times$ |
| Sparsification | $\mathcal{S}^q$ | $(d/q)^\theta$ | $(33 + \log_2 d)q$ | $0.6\times\text{–}6.0\times$ |
| **New** | $\mathcal{C}_{\mathrm{nat}} \circ \mathcal{S}^q$ | $(9d/8q)^\theta$ | $(10 + \log_2 d)q$ | $1.0\times\text{–}10.7\times$ |
| Dithering | $\mathcal{D}_{\mathrm{sta}}^{p,2^{s-1}}$ | $(1 + \kappa d^{1/r}2^{1-s})^\theta$ | $31 + d(2 + s)$ | $1.8\times\text{–}15.9\times$ |
| **New** | $\mathcal{D}_{\mathrm{nat}}^{p,s}$ | $(9/8 + \kappa d^{\frac{1}{r}}2^{1-s})^\theta$ | $31 + d(2 + \log_2 s)$ | $4.1\times\text{–}16.0\times$ |

**Table 1:** The overall speedup of distributed SGD with compression on nodes via $\mathcal{C}_{W_i}$ over a Baseline variant without compression. Speed is measured by multiplying the # communication rounds (i.e., iterations $T(\omega_W)$) by the bits sent from worker $i$ to master ($W_i \mapsto M$) per 1 iteration. We neglect $M \mapsto W_i$ communication as in practice this is often much faster (see e.g. (Mishchenko et al., 2019), for other cost/speed model see Appendix D.7). We do not just restrict to this scenario and . We assume *binary*32 representation. The relative # iterations sufficient to guarantee $\varepsilon$ optimality is $T'(\omega_W) := (\omega_W + 1)^\theta$, where $\theta \in (0, 1]$ (see Thm 6). Note that in the big $n$ regime the iteration bound $T(\omega_W)$ is better due to $\theta \approx 0$ (however, this is not very practical as $n$ is usually small), while for small $n$ we have $\theta \approx 1$. For dithering, $r = \min\{p, 2\}$, $\kappa = \min\{1, \sqrt{d}2^{1-s}\}$. The lower bound for the Speedup Factor is obtained for $\theta = 1$, and the upper bound for $\theta = 0$. The Speedup Factor $\left( \frac{T(\omega_W) \cdot \#\mathrm{Bits}}{T(0) \cdot 32d} \right)$ figures were calculated for $d = 10^6$, $q = 0.1d$ (10% sparsity), $p = 2$ and optimal choice of $s$ with respect to speedup.

sense $\mathcal{D}_{\mathrm{nat}}^{p,s}$ *can be seen as a limited precision variant of* $\mathcal{C}_{\mathrm{nat}}$. As is the case with $\mathcal{C}_{\mathrm{nat}}$, the mantissa is ignored, and one communicates exponents only. The norm compression is particularly useful on the master side since multiplication by a naturally compressed norm is just summation of the exponents. The main result of this section establishes natural dithering as belonging to the class $\mathbb{B}(\omega)$:

**Theorem 4.** $\mathcal{D}_{\mathrm{nat}}^{p,s} \in \mathbb{B}(\omega)$, *where* $\omega = 1/8 + d^{1/r}2^{1-s} \min\left\{1, d^{1/r}2^{1-s}\right\}$, *and* $r = \min\{p, 2\}$.

To illustrate the strength of this result, we now compare natural dithering $\mathcal{D}_{\mathrm{nat}}^{p,s}$ to standard dithering $\mathcal{D}_{\mathrm{sta}}^{p,s}$ and show that *natural dithering is exponentially better than standard dithering*. In particular, for the same level of variance, $\mathcal{D}_{\mathrm{nat}}^{p,s}$ uses only $s$ levels while $\mathcal{D}_{\mathrm{sta}}^{p,u}$ uses $u = 2^{s-1}$ levels. Note also that the levels used by $\mathcal{D}_{\mathrm{nat}}^{p,s}$ form a *subset* of the levels used by $\mathcal{D}_{\mathrm{sta}}^{p,s}$ (see Fig 22). We also confirm this empirically (see Appendix A.4).

**Theorem 5.** *Fixing $s$, natural dithering $\mathcal{D}_{\mathrm{nat}}^{p,s}$ has $\mathcal{O}(2^{s-1}/s)$ times smaller variance than standard dithering $\mathcal{D}_{\mathrm{sta}}^{p,s}$. Fixing $\omega$, if $u = 2^{s-1}$, then $\mathcal{D}_{\mathrm{sta}}^{p,u} \in \mathbb{B}(\omega)$ implies that $\mathcal{D}_{\mathrm{nat}}^{p,s} \in \mathbb{B}(9/8(\omega + 1) - 1)$.*

## 4 DISTRIBUTED SGD

There are several stochastic gradient-type methods (Robbins & Monro, 1951; Bubeck et al., 2015; Ghadimi & Lan, 2013; Mishchenko et al., 2019) for solving (1) that are compatible with compression operators $\mathcal{C} \in \mathbb{B}(\omega)$, and hence also with our natural compression ($\mathcal{C}_{\mathrm{nat}}$) and natural dithering ($\mathcal{D}_{\mathrm{nat}}^{p,s}$) techniques. However, as none of them support compression at the master node we propose a distributed SGD algorithm that allows for *bidirectional compression* (Algorithm 1 in Appendix D.1). We note that there are two concurrent papers to ours (all appeared online in the same month and year) proposing the use of bidirectional compression, albeit in conjunction with different underlying algorithms, such as SGD with error feedback or local updates (Tang et al., 2019; Zheng et al., 2019). Since we instead focus on vanilla distributed SGD with bidirectional compression, the algorithmic part of our paper is complementary to theirs. Moreover, our key contribution—the highly efficient natural compression and dithering compressors—can be used within their algorithms as well, which expands their impact further.

We assume repeated access to unbiased stochastic gradients $g_i(x^k)$ with bounded variance $\sigma_i^2$ for every worker $i$. We also assume *node similarity* represented by constant $\zeta_i^2$, and that $f$ is $L$-smooth (gradient is $L$-Lipschitz). Formal definitions as well as detailed explanation of Algorithm 1 can be found in Appendix D. We denote $\zeta^2 = \frac{1}{n}\sum_{i=1}^n \zeta_i^2$, $\sigma^2 = \frac{1}{n}\sum_{i=1}^n \sigma_i^2$ and

$$\alpha = \frac{(\omega_M+1)(\omega_W+1)}{n}\sigma^2 + \frac{(\omega_M+1)\omega_W}{n}\zeta^2, \quad \beta = 1 + \omega_M + \frac{(\omega_M+1)\omega_W}{n}, \tag{4}$$

where $\mathcal{C}_M \in \mathbb{B}(\omega_M)$ is the compression operator used by the master node, $\mathcal{C}_{W_i} \in \mathbb{B}(\omega_{W_i})$ are the compression operators used by the workers and $\omega_W := \max_{i\in[n]} \omega_{W_i}$.

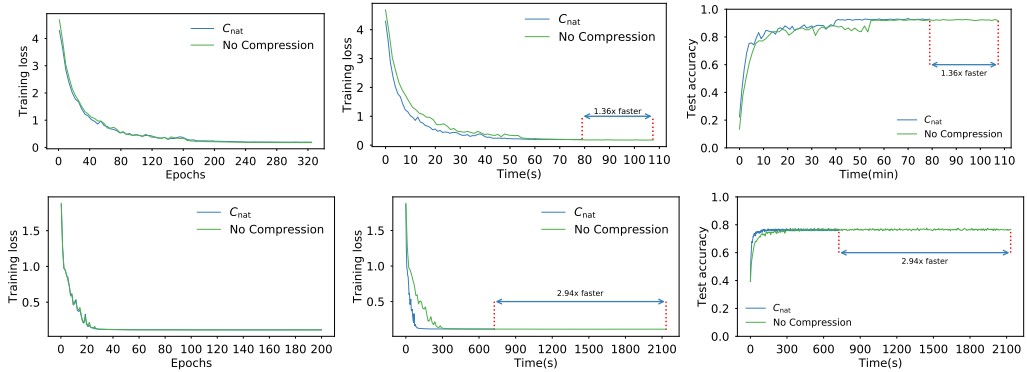

**Figure 5:** Train Loss and Test Accuracy of ResNet110 and Alexnet on CIFAR10. Speed-up is displayed with respect to time to execute fixed number of epochs, 320 and 200, respectively.

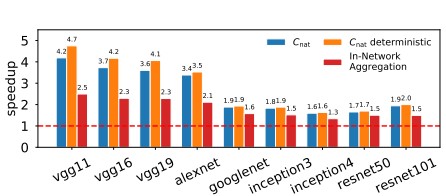

**Figure 6:** Training throughput speedup.

**Figure 7:** Accumulated transmission size of 1 worker (CIFAR10 on AlexNet).

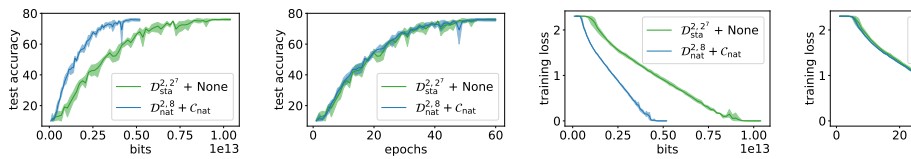

**Figure 8:** Train loss and test Aacuracy of VGG11 on CIFAR10. Green line: $\mathcal{C}_{W_i} = \mathcal{D}_{\text{sta}}^{2,2^7}$, $\mathcal{C}_M =$ identity. Blue line: $\mathcal{C}_{W_i} = \mathcal{D}_{\text{nat}}^{2,8}$, $\mathcal{C}_M = \mathcal{C}_{\text{nat}}$.

**Theorem 6.** *Let $\mathcal{C}_M \in \mathbb{B}(\omega_M)$, $\mathcal{C}_{W_i} \in \mathbb{B}(\omega_{W_i})$ and $\eta^k \equiv \eta \in (0, {}^2/_{\beta L})$, where $\alpha, \beta$ are as in (4). If $a$ is picked uniformly at random from $\{0, 1, \cdots, T-1\}$, then*

$$\mathrm{E}\left[\|\nabla f(x^a)\|^2\right] \leq \frac{2(f(x^0)-f(x^\star))}{\eta(2-\beta L\eta)T} + \frac{\alpha L\eta}{2-\beta L\eta}, \tag{5}$$

*where $x^\star$ is an opt. solution of (1). In particular, if we fix any $\varepsilon > 0$ and choose $\eta = \frac{\epsilon}{L(\alpha+\varepsilon\beta)}$ and $T \geq {}^{2L(f(x^0)-f(x^\star))(\alpha+\epsilon\beta)}/_{\varepsilon^2}$, then $\mathrm{E}\left[\|\nabla f(x^a)\|^2\right] \leq \varepsilon$.*

The above theorem has some interesting consequences. First, notice that (5) posits a $\mathcal{O}(1/T)$ convergence of the gradient norm to the value $\frac{\alpha L\eta}{2-\beta L\eta}$, which depends linearly on $\alpha$. In view of (4), the more compression we perform, the larger this value becomes. More interestingly, assume now that the same compression operator is used at each worker: $\mathcal{C}_W = \mathcal{C}_{W_i}$. Let $\mathcal{C}_W \in \mathbb{B}(\omega_W)$ and $\mathcal{C}_M \in \mathbb{B}(\omega_M)$ be the compression on master side. Then, $T(\omega_M, \omega_W) := 2L(f(x^0)-f(x^\star))\varepsilon^{-2}(\alpha+\varepsilon\beta)$ is its iteration complexity. In the special case of equal data on all nodes, i.e., $\zeta = 0$, we get $\alpha = (\omega_M+1)(\omega_W+1)\sigma^2/n$ and $\beta = (\omega_M+1)(1+\omega_W/n)$. If no compression is used, then $\omega_W = \omega_M = 0$ and $\alpha+\varepsilon\beta = \sigma^2/n+\varepsilon$. So, the *relative slowdown* of Algorithm 1 used *with* compression compared to Algorithm 1 used *without* compression is given by

$$\frac{T(\omega_M,\omega_W)}{T(0,0)} = (\omega_M + 1)\left(\frac{(\omega_W+1)\sigma^2}{n} + (1+\omega_W/n)\varepsilon\right)\Big/\left(\sigma^2/n+\varepsilon\right) \in \left(\omega_M + 1, (\omega_M+1)(\omega_W+1)\right].$$

The upper bound is achieved for $n = 1$ (or for any $n$ and $\varepsilon \to 0$), and the lower bound is achieved in the limit as $n \to \infty$. So, *the slowdown caused by compression on worker side decreases with*

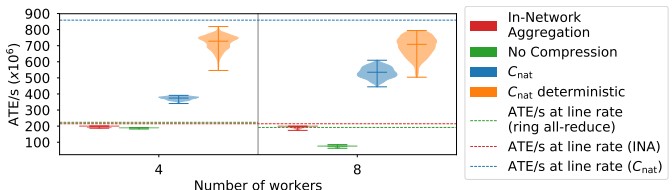
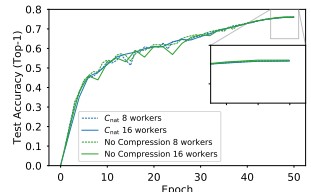

**Figure 9:** Violin plot of Aggregated Tensor Elements (ATE) per second. Dashed lines denote the maximum ATE/s under line rate.

**Figure 10:** Convergence comparison for weak scaling.

$n$. More importantly, *the savings in communication due to compression can outweigh the iteration slowdown, which leads to an overall speedup!* See Table 1 for the computation of the overall worker to master speedup achieved by our compression techniques (also see Appendix D.7 for additional similar comparisons under different cost/speed models). Notice that, however, standard sparsification does *not* necessarily improve the overall running time — it can make it worse. Our methods have the desirable property of significantly uplifting the minimal speedup comparing to their "non-natural" version. The minimal speedup is more important as usually the number of nodes $n$ is not very big.

## 5 EXPERIMENTS

To showcase properties of our approach in practice, we built a proof-of-concept system and provide evaluation results. We focus on illustrating convergence behavior, training throughput improvement, and transmitted data reduction. Experimental setup is presented in Appendix B.

**Results.** We first elaborate the microbenchmark experiments of aggregated tensor elements (ATE) per second. We collect time measurements for aggregating 200 tensors with the size of 100MB, and present violin plots which show the median, min, and max values among workers. Fig 9 shows the result where we vary the number of workers between 4 and 8. The performance difference observed for the case of $\mathcal{C}_{\mathrm{nat}}$, along with the similar performance for $\mathcal{C}_{\mathrm{nat}}$ deterministic indicate that the overhead of doing stochastic rounding at the aggregator is a bottleneck.

We then illustrate the convergence behavior by training ResNet110 and AlexNet models on CIFAR10. Fig 5 shows the train loss and test accuracy over time. We note that natural compression lowers training time by $\sim 26\%$ for ResNet110 (17% more than QSGD for the same setup, see Alistarh et al. (2017) (Table 1)) and 66% for AlexNet, compared to using no compression, while the accuracy matches the results in (He et al., 2016) without any loss of final accuracy with the same hyperparameters setting, while training loss is not affected by compression. In addition, combining $\mathcal{C}_{\mathrm{nat}}$ with other compression operators, we can see no effect in convergence, but significant reduction in communication, e.g., $16\times$ fewer levels for $\mathcal{D}_{\mathrm{nat}}^{p,s}$ w.r.t. $\mathcal{D}_{\mathrm{sta}}^{p,s}$; see Fig 8 and for other compressions see Fig 19 and 20 in Appendix.

Next, we report the speedup measured in average training throughput while training benchmark CNN models on Imagenet dataset for one epoch. The throughput is calculated as the total number of images processed divided by the time elapsed. Fig 6 shows the speedup normalized by the training throughput of the baseline, that is, TensorFlow + Horovod using the NCCL communication library. We further break down the speedup by showing the relative speedup of In-Network Aggregation, which performs no compression but reduces the volume of data transferred (shown below). We also show the effects of deterministic rounding on throughput. Because deterministic rounding does not compute random numbers, it provides some additional speedups. However, it may affect convergence. These results represent potential speedups in case the overheads of randomization were low, for instance, when using simply lookup for pre-computed randomness. We observe that the *communication-intensive* models (VGG, AlexNet) benefit more from quantization as compared to the *computation-intensive* models (GoogleNet, Inception, ResNet). These observations are consistent with prior work (Alistarh et al., 2017). To quantify the data reduction benefits of natural compression, we measure the total volume of data transferred during training. Fig 7 shows that data transferred grows linearly over time, as expected. Natural compression saves 84% of data, which greatly reduces communication time. Fig 10 studies weak scaling for training ResNet50 on ImageNet showing that $\mathcal{C}_{\mathrm{nat}}$ in itself does not have a negative effect on weak scaling. Further details and additional experiments including convergence experiments for Neural Collaborative Filtering (He et al., 2017) are presented in Appendix A.

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

# Appendix

For easy navigation through the Paper and the Appendices, we provide a table of contents.

## CONTENTS

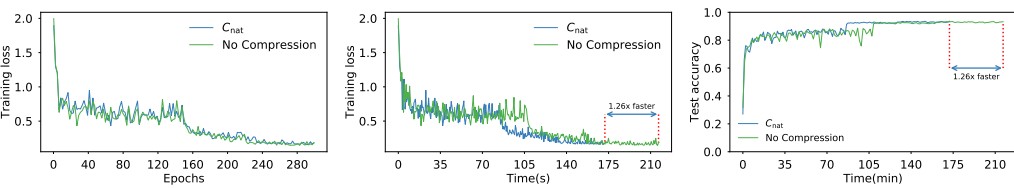

**Figure 11:** DenseNet40 ($k = 12$)

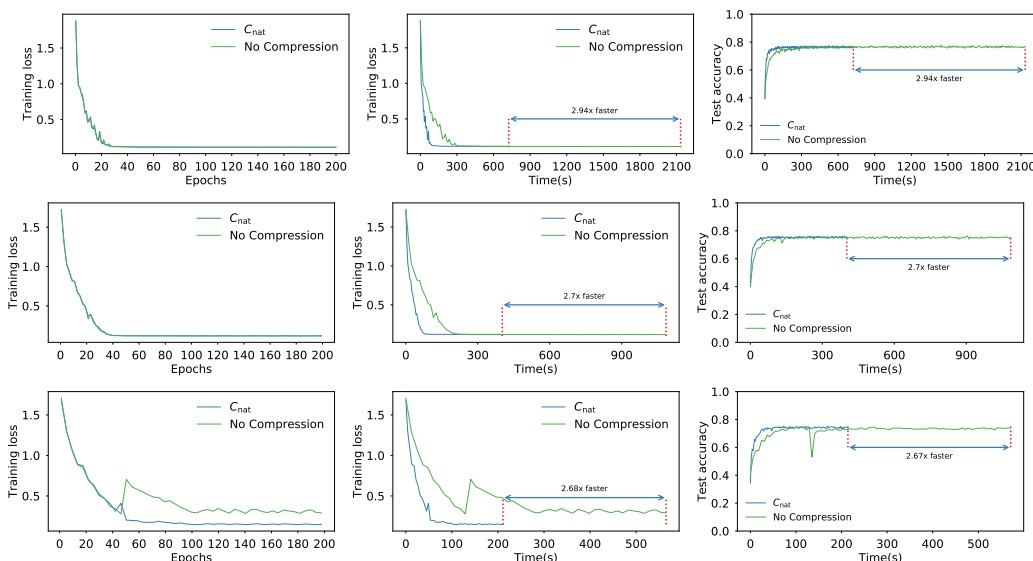

**Figure 12:** AlexNet (Batch size: 256, 512 and 1024)

# A  EXTRA EXPERIMENTS

## A.1  CONVERGENCE TESTS ON CIFAR 10

In order to validate that $\mathcal{C}_{\mathrm{nat}}$ does not incur any loss in performance, we trained various DNNs on the Tensorflow CNN Benchmark[1] on the CIFAR 10 dataset with and without $\mathcal{C}_{\mathrm{nat}}$ for the same number of epochs, and compared the test set accuracy, and training loss. As mentioned earlier, the baseline for comparison is the default NCCL setting. We didn't tune the hyperparameters. In all of the experiments, we used Batch Normalization, but no Dropout was used.

Looking into Figures 11, 12 and 13, one can see that $\mathcal{C}_{\mathrm{nat}}$ achieves significant speed-up without incurring any accuracy loss. As expected, the communication intensive AlexNet (62.5 M parameters) benefits more from the compression than the computation intensive ResNets (< 1.7 M parameters) and DenseNet40 (1 M parameters).

### A.1.1  DENSENET HYPERPARAMETERS:

We trained DenseNet40 ($k = 12$) and followed the same training procedure as described in Huang et al. (2017). We used a weight decay of $10^{-4}$ and the optimizer as vanilla SGD. We trained for a total of 300 epochs. The initial learning rate was $0.1$, which was decreased by a factor of 10 at 150 and 225 epoch.

---

[1]https://github.com/tensorflow/benchmarks

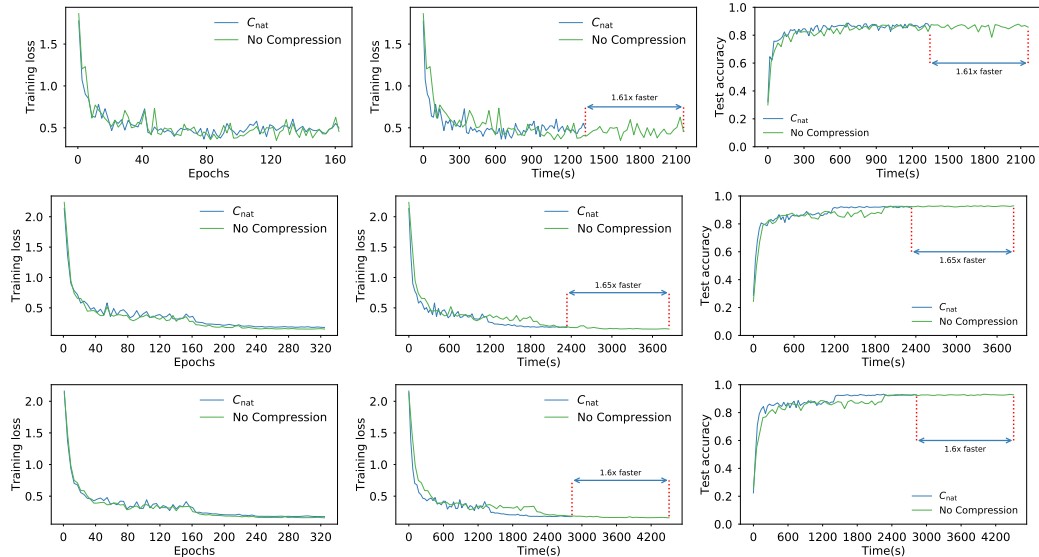

**Figure 13:** ResNet (#layers: 20, 44 and 56)

### A.1.2 ALEXNET HYPERPARAMETERS:

For AlexNet, we chose the optimizer as SGD with momentum, with a momentum of $0.9$. We trained on three minibatch sizes: $256, 512$ and $1024$ for 200 epochs. The learning rate was initially set to be $0.001$, which was decreased by a factor of $10$ after every 30 epoch.

### A.1.3 RESNET HYPERPARAMETERS:

All the ResNets followed the training procedure as described in He et al. (2016). We used a weight decay of $10^{-4}$ and the optimizer was chosen to be vanilla SGD. The minibatch size was fixed to be 128 for ResNet 20, and 256 for all the others. We train for a total of 64K iterations. We start with an initial learning rate of $0.1$, and multiply it by $0.1$ at $32K$ and $48K$ iterations.

### A.2 CONVERGENCE TESTS ON IMAGENET

To further demonstrate the convergence behavior of $\mathcal{C}_{\mathrm{nat}}$, we run experiments which conform to the ImageNet Large Scale Visual Recognition Challenge (ILSVRC). We follow publicly available benchmark[2] and apply $\mathcal{C}_{\mathrm{nat}}$ on it without modifying any hyperparameter. The model trains for $50$ epochs on 8 and 16 workers, with default ResNet50 setup: SGD optimizer with $0.875$ momentum, cosine learning rate schedule with $0.256$ initial learning rate and linear warmup during the first $8$ epochs. The weight decay is set to $^1/_{32768}$ and is not applied on Batch Norm trainable parameters. Furthermore, $0.1$ label smoothing is used. As shown in 14, $\mathcal{C}_{\mathrm{nat}}$ does not incur any accuracy loss even if applied on large distributed tasks.

### A.3 CONVERGENCE TESTS FOR NEURAL COLLABORATIVE FILTERING

We also train Neural Collaborative Filtering (NCF) (He et al., 2017) on MovieLens-20M Dataset using $\mathcal{C}_{\mathrm{nat}}$ and compare its convergence to no compression. Neural Collaborative Filtering is a big recommendation model with $\sim$32 million parameters. We use a publicly available benchmark[3] and apply $\mathcal{C}_{\mathrm{nat}}$ on it without modifying any hyperparameter. The model trains for 20 epochs on 8 workers with ADAM optimizer (Kingma & Ba, 2015) (lr= $4.5 \times 10^{-3}$, $\beta_1 = 0.25$, $\beta_2 = 0.5$), a global

---

[2]https://github.com/NVIDIA/DeepLearningExamples/tree/master/TensorFlow/
Classification/RN50v1.5
[3]https://github.com/NVIDIA/DeepLearningExamples/tree/master/PyTorch/
Recommendation/NCF

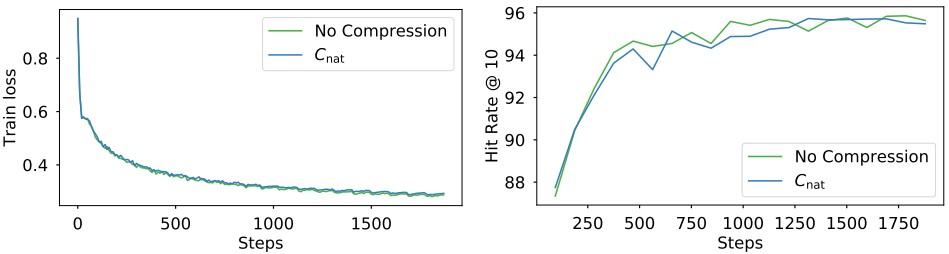

**Figure 14:** ResNet50 on ImageNet

batch-size of $2^{20}$, and a dropout ratio of $0.5$. No weigt decay is applied. Fig 15 shows that $\mathcal{C}_{\text{nat}}$ performs similar to no compression both in terms of training loss and test hit rate.

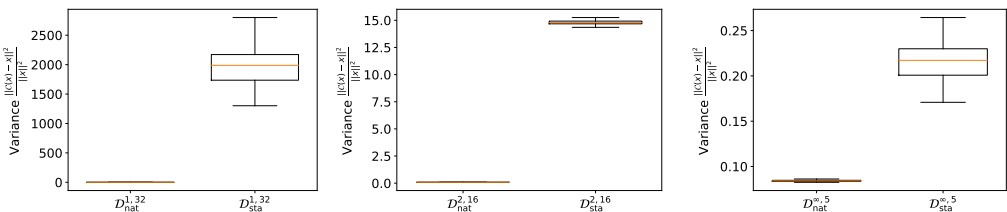

**Figure 15:** Neural Collaborative Filtering on MovieLens-20M

## A.4 $\mathcal{D}_{\text{nat}}^{p,s}$ VS. $\mathcal{D}_{\text{sta}}^{p,u}$: EMPIRICAL VARIANCE

In this section, we perform experiments to confirm that $\mathcal{D}_{\text{nat}}^{p,s}$ level selection brings not just theoretical but also practical performance speedup in comparison to $\mathcal{D}_{\text{sta}}^{p,u}$. We measure the empirical variance of $\mathcal{D}_{\text{sta}}^{p,u}$ and $\mathcal{D}_{\text{nat}}^{p,s}$. For $\mathcal{D}_{\text{nat}}^{p,s}$, we do not compress the norm, so we can compare just variance introduced by level selection. Our experimental setup is the following. We first generate a random vector $x$ of size $d = 10^5$, with independent entries with Gaussian distribution of zero mean and unit variance (we tried other distributions, the results were similar, thus we report just this one) and then we measure normalized *empirical variance*

$$\omega(x) := \frac{\|\mathcal{C}(x) - x\|^2}{\|x\|^2}.$$

We provide boxplots, each for 100 randomly generated vectors $x$ using the above procedure. We perform this for $p = 1$, $p = 2$ and $p = \infty$. We report our findings in Fig 16, Fig 17 and Fig 18. These experimental results support our theoretical findings.

### A.4.1 $\mathcal{D}_{\text{nat}}^{p,s}$ HAS EXPONENTIALLY BETTER VARIANCE

In Fig 16, we compare $\mathcal{D}_{\text{nat}}^{p,s}$ and $\mathcal{D}_{\text{sta}}^{p,u}$ for $u = s$, i.e., we use the same number of levels for both compression strategies. In each of the three plots we generated vectors $x$ with a different norm. We find that natural dithering has dramatically smaller variance, as predicted by Thm 5.

**Figure 16:** $\mathcal{D}_{\text{nat}}^{p,s}$ vs. $\mathcal{D}_{\text{sta}}^{p,u}$ with $u = s$.

### A.4.2 $\mathcal{D}_{\mathrm{nat}}^{p,s}$ NEEDS EXPONENTIALLY LESS LEVELS TO ACHIEVE THE SAME VARIANCE

In Fig 17, we set the number of levels for $\mathcal{D}_{\mathrm{sta}}^{p,u}$ to $u = 2^{s-1}$. That is, we give standard dithering an exponential advantage in terms of the number of levels (which also means that it will need more bits for communication). We now study the effect of this change on the variance. We observe that the empirical variance is essentially the same for both, as predicted by Thm 5.

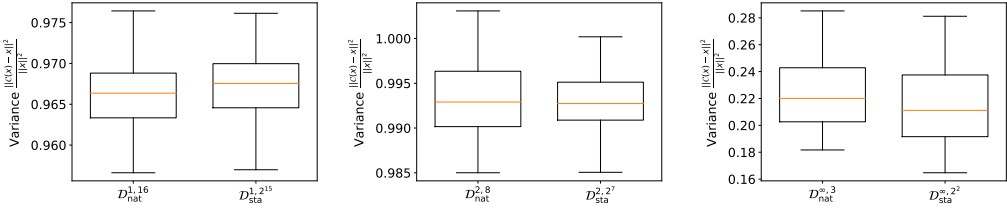

**Figure 17:** $\mathcal{D}_{\mathrm{nat}}^{p,s}$ vs. $\mathcal{D}_{\mathrm{sta}}^{p,u}$ with $u = 2^{s-1}$.

### A.4.3 $\mathcal{D}_{\mathrm{sta}}^{p,s}$ CAN OUTPERFORM $\mathcal{D}_{\mathrm{nat}}^{p,s}$ IN THE BIG $s$ REGIME

We now remark on the situation when the number of levels $s$ is chosen to be very large (see Fig 18). While this is not a practical setting as it does not provide sufficient compression, it will serve as an illustration of a fundamental theoretical difference between $\mathcal{D}_{\mathrm{sta}}^{p,s}$ and $\mathcal{D}_{\mathrm{nat}}^{p,s}$ in the $s \to \infty$ limit which we want to highlight. Note that while $\mathcal{D}_{\mathrm{sta}}^{p,s}$ converges to the identity operator as $s \to \infty$, which enjoys zero variance, $\mathcal{D}_{\mathrm{nat}}^{p,s}$ converges to $\mathcal{C}_{\mathrm{nat}}$ instead, with variance that can't reduce below $\omega = 1/8$. Hence, for large enough $s$, one would expect, based on our theory, the variance of $\mathcal{D}_{\mathrm{nat}}^{p,s}$ to be around $1/8$, while the variance of $\mathcal{D}_{\mathrm{sta}}^{p,s}$ to be closer to zero. In particular, this means that $\mathcal{D}_{\mathrm{sta}}^{p,s}$ *can*, in a practically meaningless regime, outperform $\mathcal{D}_{\mathrm{nat}}^{p,s}$. In Fig 18 we choose $p = \infty$ and $s = 32$ (this is large). Note that, as expected, the empirical variance of both compression techniques is small, and that, indeed, $\mathcal{D}_{\mathrm{sta}}^{p,s}$ outperforms $\mathcal{D}_{\mathrm{nat}}^{p,s}$.

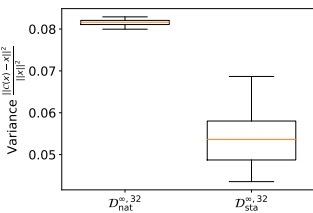

**Figure 18:** When $p = \infty$ and $s$ is very large, the empirical variance of $\mathcal{D}_{\mathrm{sta}}^{p,s}$ can be smaller than that of $\mathcal{D}_{\mathrm{nat}}^{p,s}$. However, in this case, the variance of $\mathcal{D}_{\mathrm{nat}}^{p,s}$ is already negligible.

### A.4.4 COMPRESSING GRADIENTS

We also performed identical to those reported above, but with a different generation technique of the vectors $x$. In particular, instead of a synthetic Gaussian generation, we used gradients generated by our optimization procedure as applied to the problem of training several deep learning models. Our results were essentially the same as the ones reported above, and hence we do not include them.

### A.5 DIFFERENT COMPRESSION OPERATORS

We report additional experiments where we compare our compression operator to previously proposed ones. These results are based on a Python implementation of our methods running in PyTorch as this enabled a rapid direct comparisons against the prior methods. We compare against no compression, random sparsification, and random dithering methods. We compare on MNIST and CIFAR10 datasets. For MNIST, we use a two-layer fully connected neural network with RELU activation function. For CIFAR10, we use VGG11 with one fully connected layer as the classifier. We run these experiments

with $4$ workers and batch size $32$ for MNIST and $64$ for CIFAR10. The results are averages over 3 runs.

We tune the step size for SGD for a given "non-natural" compression. Then we use the same step size for the "natural" method. Step sizes and parameters are listed alongside the results.

Figures 19 and 20 illustrate the results. Each row contains four plots that illustrate, left to right, (1) the test accuracy vs. the volume of data transmitted from workers to master (measured in bits), (2) the test accuracy over training epochs, (3) the training loss vs. the volume of data transmitted, and (4) the training loss over training epochs.

One can see that in terms of epochs, we obtain almost the same result in terms of training loss and test accuracy, sometimes even better. On the other hand, our approach has a huge impact on the number of bits transmitted from workers to master, which is the main speedup factor together with the speedup in aggregation if we use In-Network Aggregation (INA). Moreover, with INA we compress updates also from master to nodes, hence we send also fewer bits. These factors together bring significant speedup improvements, as illustrated in Fig 6, which **strongly suggests** similar **speed-up in training time** as observed for $\mathcal{C}_{\text{nat}}$, see e.g. Section 5.

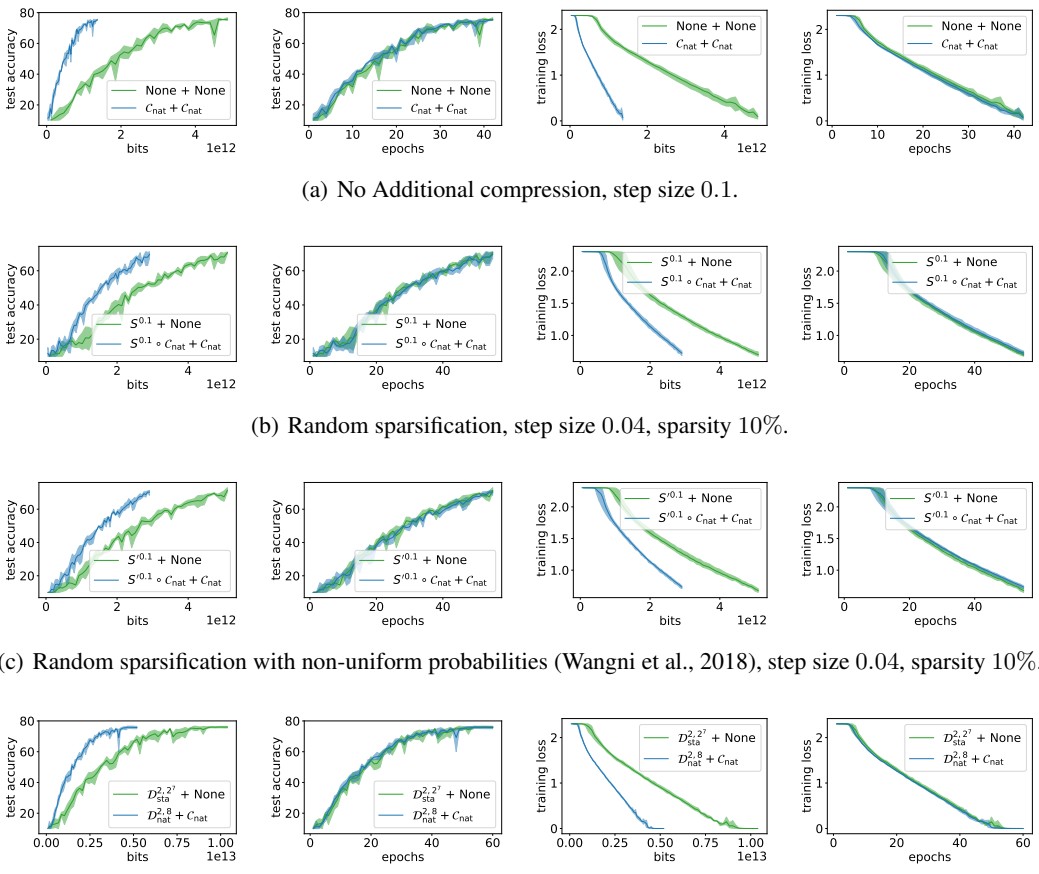

(a) No Additional compression, step size 0.1.

(b) Random sparsification, step size 0.04, sparsity 10%.

(c) Random sparsification with non-uniform probabilities (Wangni et al., 2018), step size 0.04, sparsity 10%.

(d) Random dithering, step size 0.08, $s = 8$, $u = 2^7$, second norm.

**Figure 19:** CIFAR10 with VGG11.

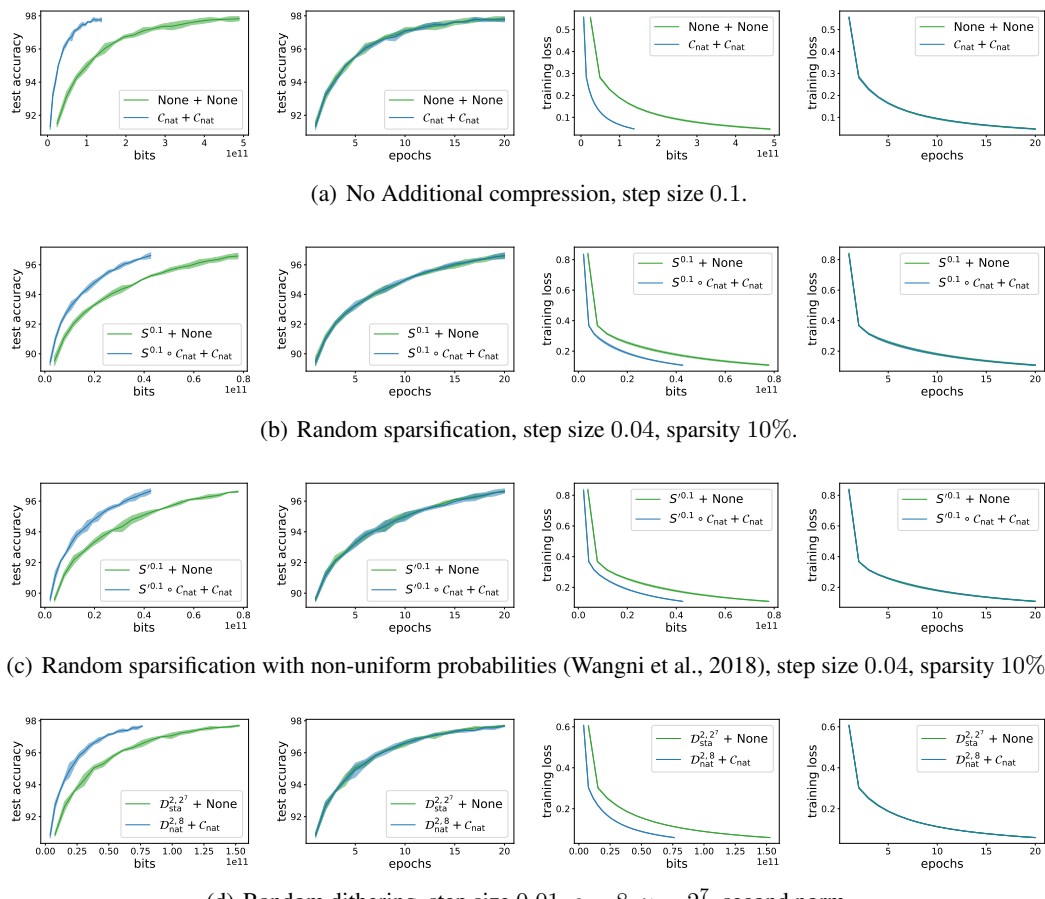

(a) No Additional compression, step size 0.1.

(b) Random sparsification, step size 0.04, sparsity 10%.

(c) Random sparsification with non-uniform probabilities (Wangni et al., 2018), step size 0.04, sparsity 10%.

(d) Random dithering, step size 0.01, $s = 8$, $u = 2^7$, second norm.

**Figure 20:** MNIST with 2 fully conected layers.

## B EXPERIMENTAL SETUP

Our experiments execute the standard CNN benchmark[4]. We summarize the hyperparameters setting in Appendix A.1.2. We further present results for two more variations of our implementation: one without compression (providing the baseline for In-Network Aggregation (Sapio et al., 2019)), and the other with deterministic rounding to the nearest power of 2 to emphasize that there exists a performance overhead of sampling in natural compression.

We implement the natural compression operator within the Gloo communication library[5], as a drop-in replacement for the ring all-reduce routine. Our implementation is in C++. We integrate our communication library with Horovod and, in turn, with TensorFlow. We follow the same communication strategy introduced in SwitchML (Sapio et al., 2019), which aggregates the deep learning model's gradients using In-Network Aggregation on programmable network switches. We choose this strategy because natural compression is a good fit for the capabilities of this class of modern hardware, which only supports basic integer arithmetic, simple logical operations and limited storage.

A worker applies the natural compression operator to quantize gradient values and sends them to the aggregator component. As in SwitchML, an aggregator is capable of aggregating a fixed-length array of gradient values at a time. Thus, the worker sends a stream of network packets, each carrying a chunk of compressed values. For a given chunk, the aggregator awaits all values from every worker;

---

[4]https://github.com/tensorflow/benchmarks
[5]https://github.com/facebookincubator/gloo

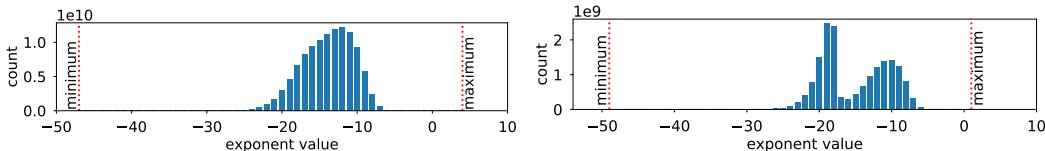

**Figure 21:** Histogram of exponents of gradients exchanged during the entire training process for ResNet110 (left) and Alexnet (right). Red lines denote the minimum and maximum exponent values of all gradients.

then, it restores the compressed values as integers, aggregates them and applies compression to quantize the aggregated values. Finally, the aggregator multicasts back to the workers a packet of aggregated values.

For implementation expedience, we prototype the In-Network Aggregation as a server-based program implemented atop DPDK[6] for fast I/O performance. We leave to future work a complete P4 implementation for programmable switches; however, we note that all operations (bit shifting, masking, and random bits generation) needed for our compression operator are available on programmable switches.

**Implementation optimization.** We carefully optimize our implementation using modern x86 vector instructions (AVX2) to minimize the overheads in doing compression. To fit the byte length and access memory more efficiently, we compress a 32-bit floating point numbers to an 8-bit representation, where 1 bit is for the sign and 7 bits are for the exponent. The aggregator uses 64-bit integers to store the intermediate results, and we choose to clip the exponents in the range of $-50 \sim 10$. As a result, we only use 6 bits for exponents. The remaining one bit is used to represent zeros. Note that it is possible to implement 128-bit integers using two 64-bit integers, but we found that, in practice, the exponent values never exceed the range of $-50 \sim 10$ (Figure 21).

Despite the optimization effort, we identify non-negligible $10 \sim 15\%$ overheads in doing random number generation used in stochastic rounding, which was also reported in Hubara et al. (2017). We include the experimental results of our compression operator without stochastic rounding as a reference. There could be more efficient ways to deal with stochastic rounding, but we observe that doing deterministic rounding gives nearly the same training curve in practice meaning that computational speed up is neutralized by slower convergence due to biased compression operator.

**Hardware setup.** We run the workers on 8 machines configured with 1 NVIDIA P100 GPU, dual CPU Intel Xeon E5-2630 v4 at 2.20GHz, and 128 GB of RAM. The machines run Ubuntu (Linux kernel 4.4.0-122) and CUDA 9.0. Following Sapio et al. (2019), we balance the workers with 8 aggregators (4 aggregators in the case of 4 workers) running on machines configured with dual Intel Xeon Silver 4108 CPU at 1.80 GHz. Each machine uses a 10 GbE network interface and has CPU frequency scaling disabled. The chunks of compressed gradients sent by workers are uniformly distributed across all aggregators. This setup ensures that workers can fully utilize their network bandwidth and match the performance of a programmable switch. We leave the switch-based implementation for future work.

---

[6] https://www.dpdk.org

## C  Details and Proofs for Sections 2 and 3

### C.1  Proof of Theorem 1

By linearity of expectation, the unbiasedness condition and the second moment condition (3) have the form

$$\mathrm{E}\left[(\mathcal{C}(x))_i\right] = x_i, \qquad \forall x \in \mathbb{R}^d, \quad \forall i \in [d] \tag{6}$$

and

$$\sum_{i=1}^d \mathrm{E}\left[(\mathcal{C}(x))_i^2\right] \le (\omega + 1)\sum_{i=1}^d x_i^2, \qquad \forall x \in \mathbb{R}^d. \tag{7}$$

Recall that $\mathcal{C}_{\mathrm{nat}}(t)$ can be written in the form

$$\mathcal{C}_{\mathrm{nat}}(t) = \mathrm{sign}(t) \cdot 2^{\lfloor \log_2 |t| \rfloor}(1 + \lambda(t)). \tag{8}$$

where the last step follows since $p(t) = \frac{2^{\lceil \log_2 |t| \rceil} - |t|}{2^{\lfloor \log_2 |t| \rfloor}}$. Hence,

$$\mathrm{E}\left[\mathcal{C}_{\mathrm{nat}}(t)\right] \overset{(8)}{=} \mathrm{E}\left[\mathrm{sign}(t) \cdot 2^{\lfloor \log_2 |t| \rfloor}(1 + \lambda(t))\right] = \mathrm{sign}(t) \cdot 2^{\lfloor \log_2 |t| \rfloor}\left(1 + \mathrm{E}\left[\lambda(t)\right]\right)$$

$$= \mathrm{sign}(t) \cdot 2^{\lfloor \log_2 |t| \rfloor}\left(1 + 1 - p(t)\right) = t,$$

This establishes unbiasedness (6).

In order to establish (7), it suffices to show that $\mathrm{E}\left[(\mathcal{C}_{\mathrm{nat}}(x))_i^2\right] \le (\omega + 1)x_i^2$ for all $x_i \in \mathbb{R}$. Since by definition $(\mathcal{C}_{\mathrm{nat}}(x))_i = \mathcal{C}_{\mathrm{nat}}(x_i)$ for all $i \in [d]$, it suffices to show that

$$\mathrm{E}\left[(\mathcal{C}_{\mathrm{nat}}(t))^2\right] \le (\omega + 1)t^2, \qquad \forall t \in \mathbb{R}. \tag{9}$$

If $t = 0$ or $t = \mathrm{sign}(t)2^\alpha$ with $\alpha$ being an integer, then $\mathcal{C}_{\mathrm{nat}}(t) = t$, and (9) holds as an identity with $\omega = 0$, and hence inequality (9) holds for $\omega = 1/8$. Otherwise $t = \mathrm{sign}(t)2^\alpha$ where $a := \lfloor \alpha \rfloor < \alpha < \lceil \alpha \rceil = a + 1$. With this notation, we can write

$$\mathrm{E}\left[(\mathcal{C}_{\mathrm{nat}}(t))^2\right] = 2^{2a}\frac{2^{a+1} - |t|}{2^a} + 2^{2(a+1)}\frac{|t| - 2^a}{2^a} = 2^a(3|t| - 2^{a+1}).$$

So,

$$\frac{\mathrm{E}\left[(\mathcal{C}_{\mathrm{nat}}(t))^2\right]}{t^2} = \frac{2^a(3|t| - 2^{a+1})}{t^2} \le \sup_{2^a < t < 2^{a+1}} \frac{2^a(3|t| - 2^{a+1})}{t^2}$$

$$= \sup_{1 < \theta < 2} \frac{2^a(3 \cdot 2^a\theta - 2^{a+1})}{(2^a\theta)^2} = \sup_{1 < \theta < 2} \frac{3\theta - 2}{\theta^2}.$$

The optimal solution of the last maximization problem is $\theta = \frac{4}{3}$, with optimal objective value $\frac{9}{8}$. This implies that (9) holds with $\omega = \frac{1}{8}$.

### C.2  Proof of Theorem 2

Let assume that there exists some $\omega < \infty$ for which $\mathcal{C}_{\mathrm{int}}$ is the $\omega$ quantization. Unbiased rounding to the nearest integer can be defined in the following way

$$\mathcal{C}_{\mathrm{int}}(x_i) := \begin{cases} \lfloor x_i \rfloor, & \text{with probability} \quad p(x_i), \\ \lceil x_i \rceil, & \text{with probability} \quad 1 - p(x_i), \end{cases}$$

where $p(x_i) = \lceil x_i \rceil - x_i$. Let's take 1-D example, where $x \in (0, 1)$, then

$$\mathrm{E}\left[\mathcal{C}_{\mathrm{int}}(x^2)\right] = (1 - x)0^2 + x1^2 = x \le \omega x^2,$$

which implies $\omega \ge 1/x$, thus taking $x \to 0^+$, one obtains $\omega \to \infty$, which contradicts the existence of finite $\omega$.

### C.3 PROOF OF THEOREM 3

The main building block of the proof is the tower property of mathematical expectation. The tower property says: If $X$ and $Y$ are random variables, then $\mathrm{E}[X] = \mathrm{E}[\mathrm{E}[X \mid Y]]$. Applying it to the composite compression operator $\mathcal{C}_1 \circ \mathcal{C}_2$, we get

$$\mathrm{E}[(\mathcal{C}_1 \circ \mathcal{C}_2)(x)] = \mathrm{E}[\mathrm{E}[\mathcal{C}_1(\mathcal{C}_2(x)) \mid \mathcal{C}_2(x)]] \overset{(3)}{=} \mathrm{E}[\mathcal{C}_2(x)] \overset{(3)}{=} x \ .$$

For the second moment, we have

$$
\begin{aligned}
\mathrm{E}\left[\|(\mathcal{C}_1 \circ \mathcal{C}_2)(x)\|^2\right] &= \mathrm{E}\left[\mathrm{E}\left[\|\mathcal{C}_1(\mathcal{C}_2(x))\|^2 \mid \mathcal{C}_2(x)\right]\right] \\
&\overset{(3)}{\leq} (\omega_2 + 1)\mathrm{E}\left[\|\mathcal{C}_1(x)\|^2\right] \\
&\overset{(3)}{\leq} (\omega_1 + 1)(\omega_2 + 1)\|x\|^2 \ ,
\end{aligned}
$$

which concludes the proof.

### C.4 PROOF OF THEOREM 4

Unbiasedness of $\mathcal{D}_{\mathrm{nat}}^{p,s}$ is a direct consequence of unbiasedness of $\mathcal{D}_{\mathrm{gen}}^{\mathcal{C},p,s}$.

For the second part, we first establish a bound on the second moment of $\xi$:

$$
\begin{aligned}
\mathrm{E}\left[\xi\left(\frac{x_i}{\|x\|_p}\right)^2\right] &\leq \mathbb{1}\left(\frac{|x_i|}{\|x\|_p} \geq 2^{1-s}\right)\frac{9}{8}\frac{|x_i|^2}{\|x\|_p^2} + \mathbb{1}\left(\frac{|x_i|}{\|x\|_p} < 2^{1-s}\right)\frac{|x_i|}{\|x\|_p}2^{1-s} \\
&\leq \frac{9}{8}\frac{|x_i|^2}{\|x\|_p^2} + \mathbb{1}\left(\frac{|x_i|}{\|x\|_p} < 2^{1-s}\right)\frac{|x_i|}{\|x\|_p}2^{1-s} \ .
\end{aligned}
$$

Using this bound, we have

$$
\begin{aligned}
\mathrm{E}\left[\|\mathcal{D}_{\mathrm{nat}}^{p,s}(x)\|^2\right] &= \mathrm{E}\left[\|x\|_p^2\right]\sum_{i=1}^{d}\mathrm{E}\left[\xi\left(\frac{x_i}{\|x\|_p}\right)^2\right] \\
&\overset{(10)}{\leq} \|x\|_p^2\left(\frac{9\|x\|^2}{8\|x\|_p^2} + \sum_{i=1}^{d}\mathbb{1}\left(\frac{|x_i|}{\|x\|_p} < 2^{1-s}\right)\frac{|x_i|}{\|x\|_p}2^{1-s}\right) \\
&\leq \frac{9}{8}\|x\|^2 + \min\left\{2^{1-s}\|x\|_p\|x\|_1, 2^{2-2s}d\|x\|_p^2\right\} \\
&\leq \frac{9}{8}\|x\|^2 + \min\left\{d^{1/2}2^{1-s}\|x\|_p\|x\|, 2^{2-2s}d\|x\|_p^2\right\} \\
&\leq \left(\frac{9}{8} + d^{1/\min\{p,2\}}2^{1-s}\min\left\{1, d^{1/\min\{p,2\}}2^{1-s}\right\}\right)\|x\|^2 \ ,
\end{aligned}
$$

where the second inequality follows from $\sum\min\{a_i, b_i\} \leq \min\{\sum a_i, \sum b_i\}$ and the last two inequalities follow from the following consequence of Hölder's inequality $\|x\|_p \leq d^{1/p-1/2}\|x\|$ for $1 \leq p < 2$ and from the fact that $\|x\|_p \leq \|x\|$ for $p \geq 2$. This concludes the proof.

### C.5 PROOF OF THEOREM 5

The main building block of the proof is useful connection between $\mathcal{D}_{\mathrm{nat}}^{p,s}$ and $\mathcal{D}_{\mathrm{sta}}^{p,2^{s-1}}$, which can be formally written as

$$\mathcal{D}_{\mathrm{nat}}^{p,s}(x) \overset{D}{=} \|x\|_p \cdot \mathrm{sign}(x) \cdot \mathcal{C}_{\mathrm{nat}}(\xi(x)) \ , \tag{10}$$

where $(\xi(x))_i = \xi(x_i/\|x\|_p)$ with levels $0, 1/2^{s-1}, 2/2^{s-1}, \cdots, 1$. Graphical visualization can be found in Fig 22.

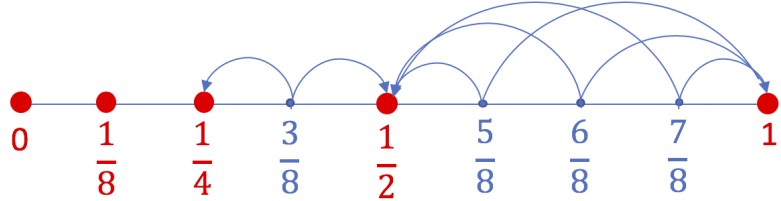

**Figure 22:** 1D visualization of the workings of natural dithering $\mathcal{D}_{\text{nat}}^{p,s}$ and standard dithering $\mathcal{D}_{\text{sta}}^{p,u}$ with $u = 2^{s-1}$, with $s = 4$. Notice that the numbers standard dithering rounds to, i.e., $0, 1/8, 2/8, \ldots, 7/8, 1$, form a *superset* of the numbers natural dithering rounds to, i.e., $0, 2^{-3}, 2^{-2}, 2^{-1}, 1$. Importantly, while standard dithering uses $u = 2^{4-1} = 8$ levels (i.e., intervals) to achieve a certain fixed variance, natural dithering only needs $s = 4$ levels to achieve the same variance. This is an exponential improvement in compression (see Theorem 5 for the formal statement).

Equipped with this, we can proceed with

$$
\begin{aligned}
\mathrm{E}\left[\left\|\xi(x_i/\|x\|_p)\right\|^2\right] & \overset{(10)}{=} & \mathrm{E}\left[\left\|\|x\|_p \cdot \operatorname{sign}(x) \cdot \mathcal{C}_{\text{nat}}(\xi(x))\right\|^2\right] \\
& = & \mathrm{E}\left[\|x\|_p^2\right] \cdot \mathrm{E}\left[\|\mathcal{C}_{\text{nat}}(\xi(x))\|^2\right] \\
& \overset{\text{Thm. 1}}{\leq} & \frac{9}{8}\mathrm{E}\left[\left\|\|x\|_p \operatorname{sign}(x)\xi(x)\right\|^2\right] \\
& = & \frac{9}{8}\mathrm{E}\left[\left\|\mathcal{D}_{\text{sta}}^{p,2^{s-1}}\right\|^2(x)\right] \\
& \leq & \frac{9}{8}(\omega + 1),
\end{aligned}
$$

which concludes the proof.

### C.6 NATURAL COMPRESSION AND DITHERING ALLOW FOR FAST AGGREGATION

Besides communication savings, our new compression operators $\mathcal{C}_{\text{nat}}$ (natural compression) and $\mathcal{D}_{\text{nat}}^{p,s}$ (natural dithering) bring another advantage, which is *ease of aggregation*. Firstly, our updates allow in-network aggregation on a primitive switch, which can speed up training by up to $300\%$ (Sapio et al., 2019) itself. Moreover, our updates are so simple that if one uses integer format on the master side for update aggregation, then our updates have just one non-zero bit, which leads to additional speed up. For this reason, one needs to operate with at least $64$ bits during the aggregation step, which is the reason why we also do $\mathcal{C}_{\text{nat}}$ compression on the master side; and hence we need to transmit just exponent to workers. Moreover, the translation from floats to integers and back is computation-free due to structure of our updates. Lastly, for $\mathcal{D}_{\text{nat}}^{p,s}$ compression we obtain additional speed up with respect to standard randomized dithering $\mathcal{D}_{\text{sta}}^{p,s}$ as our levels are computationally less expensive due to their natural compatibility with floating points. In addition, for effective communication one needs to communicate signs, norm and levels as a tuple for both $\mathcal{D}_{\text{nat}}^{p,s}$ and $\mathcal{D}_{\text{sta}}^{p,s}$, which needs to be then multiplied back on the master side. For $\mathcal{D}_{\text{nat}}^{p,s}$, this is just the summation of exponents rather than actual multiplication as is the case for $\mathcal{D}_{\text{sta}}^{p,s}$.

## D DETAILS AND PROOFS FOR SECTION 4

### D.1 ALGORITHM

---

**Algorithm 1** Distributed SGD with bidirectional compression

---

**Input:** learning rates $\{\eta^k\}_{k=0}^T > 0$, initial vector $x^0$
**for** $k = 0, 1, \ldots T$ **do**
   **Parallel: Worker side**
   **for** $i = 1, \ldots, n$ **do**
      compute a stochastic gradient $g_i(x^k)$ (of $f_i$ at $x^k$)
      compress it $\Delta_i^k = \mathcal{C}_{W_i}(g_i(x^k))$
   **end for**
   **Master side**
   aggregate $\Delta^k = \sum_{i=1}^n \Delta_i^k$
   compress $g^k = \mathcal{C}_M(\Delta^k)$ and broadcast to each worker
   **Parallel: Worker side**
   **for** $i = 1, \ldots, n$ **do**
      $x^{k+1} = x^k - \frac{\eta^k}{n} g^k$
   **end for**
**end for**

---

### D.2 Assumptions and Definitions

Formal definitions of some concepts used in Section  follows:

**Definition 4.** Let $f_i : \mathbb{R}^d \to \mathbb{R}$ be fixed function. A *stochastic gradient* for $f_i$ is a random function $g_i(x)$ so that $\mathrm{E}[g_i(x)] = \nabla f_i(x)$.

In order to obtain the rate, we introduce additional assumptions on $g_i(x)$ and $\nabla f_i(x)$.

**Assumption 1** (Bounded Variance). We say the stochastic gradient has variance at most $\sigma_i^2$ if $\mathrm{E}\left[\|g_i(x) - \nabla f_i(x)\|^2\right] \leq \sigma_i^2$ for all $x \in \mathbb{R}^d$. Moreover, let $\sigma^2 = \frac{1}{n}\sum_{i=1}^n \sigma_i^2$.

**Assumption 2** (Similarity). We say the variance of gradient among nodes is at most $\zeta_i^2$ if $\|\nabla f_i(x) - \nabla f(x)\|^2 \leq \zeta_i^2$ for all $x \in \mathbb{R}^d$. Moreover, let $\zeta^2 = \frac{1}{n}\sum_{i=1}^n \zeta_i^2$.

Moreover, we assume that $f$ is $L$-smooth (gradient is $L$-Lipschitz). These are classical assumptions for non-convex SGD (Ghadimi & Lan, 2013; Jiang & Agrawal, 2018; Mishchenko et al., 2019) and comparing to some previous works (Alistarh et al., 2017), our analysis does not require bounded iterates and bounded the second moment of the stochastic gradient. Assumption 2 is automatically satisfied with $\zeta^2 = 0$ if every worker has access to the whole dataset. If one does not like Assumption 2 one can use the DIANA algorithm (Horváth et al., 2019) as a base algorithm instead of SGD, then there is no need for this assumption. For simplicity, we decide to pursue just SGD analysis and we keep Assumption 2.

### D.3 Description of Algorithm 1

Let us describe Algorithm 1. First, each worker computes its own stochastic gradient $g_i(x^k)$, this is then compressed using a compression operator $\mathcal{C}_{W_i}$ (this can be different for every node, for simplicity, one can assume that they are all the same) and send to the master node. The master node then aggregates the updates from all the workers, compress with its own operator $\mathcal{C}_M$ and broadcasts update back to the workers, which update their local copy of the solution parameter $x$.

Note that the communication of the updates can be also done in all-to-all fashion, which implicitly results in $\mathcal{C}_M$ being the identity operator. Another application, which is one of the key motivations of our natural compression and natural dithering operators, is *in-network aggregation* (Sapio et al., 2019). In this setup, the master node is a *network switch*. However, current network switches can only perform addition (not even average) of integers.

### D.4 Three Lemmas Needed for the Proof of Theorem 6

Before we proceed with the theoretical guarantees for Algorithm 1 in smooth non-convex setting, we first state three lemmas which are used to bound the variance of $g^k$ as a stochastic estimator of

the true gradient $\nabla f(x^k)$. In this sense compression at the master-node has the effect of injecting additional variance into the gradient estimator. Unlike in SGD, where stochasticity is used to speed up computation, here we use it to reduce communication.

**Lemma 7** (Tower property + Compression). *If $\mathcal{C} \in \mathbb{B}(\omega)$ and $z$ is a random vector independent of $\mathcal{C}$, then*

$$\mathrm{E}\left[\|\mathcal{C}(z) - z\|^2\right] \leq \omega \mathrm{E}\left[\|z\|^2\right]; \qquad \mathrm{E}\left[\|\mathcal{C}(z)\|^2\right] \leq (\omega + 1)\mathrm{E}\left[\|z\|^2\right]. \tag{11}$$

*Proof.* Recall from the discussion following Definition 2 that the variance of a compression operator $\mathcal{C} \in \mathbb{B}(\omega)$ can be bounded as

$$\mathrm{E}\left[\|\mathcal{C}(x) - x\|^2\right] \leq \omega \|x\|^2, \qquad \forall x \in \mathbb{R}^d.$$

Using this with $z = x$, this can be written in the form

$$\mathrm{E}\left[\|\mathcal{C}(z) - z\|^2 \mid z\right] \leq \omega \|z\|^2, \qquad \forall x \in \mathbb{R}^d, \tag{12}$$

which we can use in our argument:

$$\begin{aligned}
\mathrm{E}\left[\|\mathcal{C}(z) - z\|^2\right] &= \mathrm{E}\left[\mathrm{E}\left[\|\mathcal{C}(z) - z\|^2 \mid z\right]\right] \\
&\overset{(12)}{\leq} \mathrm{E}\left[\omega \|z\|^2\right] \\
&= \omega \mathrm{E}\left[\|z\|^2\right].
\end{aligned}$$

The second inequality can be proved exactly same way. $\qquad \square$

**Lemma 8** (Local compression variance). *Suppose $x$ is fixed, $\mathcal{C} \in \mathbb{B}(\omega)$, and $g_i(x)$ is an unbiased estimator of $\nabla f_i(x)$. Then*

$$\mathrm{E}\left[\|\mathcal{C}(g_i(x)) - \nabla f_i(x)\|^2\right] \leq (\omega + 1)\sigma_i^2 + \omega \|\nabla f_i(x)\|^2. \tag{13}$$

*Proof.*

$$\begin{aligned}
\mathrm{E}\left[\|\mathcal{C}(g_i(x)) - \nabla f_i(x)\|^2\right] &\overset{\text{Def. }4+(3)}{=} \mathrm{E}\left[\|\mathcal{C}(g_i(x)) - g_i(x)\|^2\right] + \mathrm{E}\left[\|g_i(x) - \nabla f_i(x)\|^2\right] \\
&\overset{(11)}{\leq} \omega \mathrm{E}\left[\|g_i(x)\|^2\right] + \mathrm{E}\left[\|g_i(x) - \nabla f_i(x)\|^2\right] \\
&\overset{\text{Def. }4+(3)}{=} (\omega + 1)\mathrm{E}\left[\|g_i(x) - \nabla f_i(x)\|^2\right] + \omega \|\nabla f_i(x)\|^2 \\
&\overset{\text{Assum. }1}{\leq} (\omega + 1)\sigma_i^2 + \omega \|\nabla f_i(x)\|^2.
\end{aligned}$$

$\qquad \square$

**Lemma 9** (Global compression variance). *Suppose $x$ is fixed, $\mathcal{C}_{W_i} \in \mathbb{B}(\omega_{W_i})$ for all $i$, $\mathcal{C}_M \in \mathbb{B}(\omega_M)$, and $g_i(x)$ is an unbiased estimator of $\nabla f_i(x)$ for all $i$. Then*

$$\mathrm{E}\left[\left\|\frac{1}{n}\mathcal{C}_M\left(\sum_{i=1}^n \mathcal{C}_{W_i}(g_i(x))\right)\right\|^2\right] \leq \alpha + \beta \|\nabla f(x)\|^2, \tag{14}$$

*where $\omega_W = \max_{i \in [n]} \omega_{W_i}$ and*

$$\alpha = \frac{(\omega_M + 1)(\omega_W + 1)}{n}\sigma^2 + \frac{(\omega_M + 1)\omega_W}{n}\zeta^2, \qquad \beta = 1 + \omega_M + \frac{(\omega_M + 1)\omega_W}{n}. \tag{15}$$

*Proof.* For added clarity, let us denote

$$\Delta = \sum_{i=1}^n \mathcal{C}_{W_i}(g_i(x)).$$

Using this notation, the proof proceeds as follows:

$$
\begin{aligned}
\mathrm{E}\left[\left\|\frac{1}{n}\mathcal{C}_M(\Delta)\right\|^2\right] &\overset{\text{Def. }\underline{4}+(3)}{=} \mathrm{E}\left[\left\|\frac{1}{n}\mathcal{C}_M(\Delta)-\nabla f(x)\right\|^2\right]+\|\nabla f(x)\|^2 \\[2mm]
&\overset{\text{Def. }\underline{4}+(3)}{=} \frac{1}{n^2}\mathrm{E}\left[\|\mathcal{C}_M(\Delta)-\Delta\|^2\right]+\mathrm{E}\left[\left\|\frac{1}{n}\Delta-\nabla f(x)\right\|^2\right]+\|\nabla f(x)\|^2 \\[2mm]
&\overset{(11)}{\leq} \frac{\omega_M}{n^2}\mathrm{E}\left[\|\Delta\|^2\right]+\mathrm{E}\left[\left\|\frac{1}{n}\Delta-\nabla f(x)\right\|^2\right]+\|\nabla f(x)\|^2 \\[2mm]
&\overset{\text{Def. }\underline{4}+(3)}{=} (\omega_M+1)\mathrm{E}\left[\left\|\frac{1}{n}\Delta-\nabla f(x)\right\|^2\right]+(\omega_M+1)\|\nabla f(x)\|^2 \\[2mm]
&= \frac{\omega_M+1}{n^2}\sum_{i=1}^{n}\mathrm{E}\left[\|\mathcal{C}_{W_i}(g_i(x))-\nabla f_i(x)\|^2\right]+(\omega_M+1)\|\nabla f(x)\|^2 \\[2mm]
&\overset{(13)}{\leq} \frac{(\omega_M+1)(\omega_W+1)}{n}\sigma^2+\frac{(\omega_M+1)\omega_W}{n}\frac{1}{n}\sum_{i=1}^{n}\|\nabla f_i(x)\|^2 \\
&\qquad +(\omega_M+1)\|\nabla f(x)\|^2 \\[2mm]
&= \frac{(\omega_M+1)(\omega_W+1)}{n}\sigma^2+\frac{(\omega_M+1)\omega_W}{n}\frac{1}{n}\sum_{i=1}^{n}\|\nabla f_i(x)-\nabla f(x)\|^2 \\
&\qquad +\left(1+\omega_M+\frac{(\omega_M+1)\omega_W}{n}\right)\|\nabla f(x)\|^2 \\[2mm]
&\overset{\text{Assum. }2}{\leq} \frac{(\omega_M+1)(\omega_W+1)}{n}\sigma^2+\frac{(\omega_M+1)\omega_W}{n}\zeta^2 \\
&\qquad +\left(1+\omega_M+\frac{(\omega_M+1)\omega_W}{n}\right)\|\nabla f(x)\|^2
\end{aligned}
$$

$\square$

## D.5 PROOF OF THEOREM 6

Using $L$-smoothness of $f$ and then applying Lemma 9, we get

$$
\begin{aligned}
\mathrm{E}\left[f(x^{k+1})\right] &\leq \mathrm{E}\left[f(x^k)\right]+\mathrm{E}\left[\langle\nabla f(x^k),x^{k+1}-x^k\rangle\right]+\frac{L}{2}\mathrm{E}\left[\|x^{k+1}-x^k\|^2\right] \\[2mm]
&\leq \mathrm{E}\left[f(x^k)\right]-\eta_k\mathrm{E}\left[\|\nabla f(x^k)\|^2\right]+\frac{L}{2}\eta_k^2\mathrm{E}\left[\left\|\frac{g^k}{n}\right\|^2\right] \\[2mm]
&\overset{(14)}{\leq} \mathrm{E}\left[f(x^k)\right]-\left(\eta_k-\frac{L}{2}\beta\eta_k^2\right)\mathrm{E}\left[\|\nabla f(x^k)\|^2\right]+\frac{L}{2}\alpha\eta_k^2.
\end{aligned}
$$

Summing these inequalities for $k=0,...,T-1$, we obtain

$$
\sum_{k=0}^{T-1}\left(\eta_k-\frac{L}{2}\beta\eta_k^2\right)\mathrm{E}\left[\|\nabla f(x^k)\|^2\right]\leq f(x^0)-f(x^\star)+\frac{TL\alpha\eta_k^2}{2}.
$$

Taking $\eta_k=\eta$ and assuming

$$
\eta<\frac{2}{L\beta}, \tag{16}
$$

one obtains

$$
\mathrm{E}\left[\|\nabla f(x^a)\|^2\right]\leq\frac{1}{T}\sum_{k=0}^{T-1}\mathrm{E}\left[\|\nabla f(x^k)\|^2\right]\leq\frac{2(f(x^0)-f(x^\star))}{T\eta(2-L\beta\eta)}+\frac{L\alpha\eta}{2-L\beta\eta}:=\delta(\eta,T).
$$

It is easy to check that if we choose $\eta=\frac{\varepsilon}{L(\alpha+\varepsilon\beta)}$ (which satisfies (16) for every $\varepsilon>0$), then for any $T\geq\frac{2L(f(x^0)-f(x^\star))(\alpha+\epsilon\beta)}{\epsilon^2}$ we have $\delta(\eta,T)\leq\varepsilon$, concluding the proof.

| | Master can aggregate real numbers (e.g., a workstation) | Master can aggregate integers only (e.g., SwitchML (Sapio et al., 2019)) |
|---|---|---|
| Same communication speed both ways | MODEL 1 | MODEL 3 |
| Master communicates infinitely fast | MODEL 2 | MODEL 4 |

**Table 2:** Four theoretical models.

### D.6 A DIFFERENT STEPSIZE RULE FOR THEOREM 6

Looking at Theorem 6, one can see that setting step size

$$\eta_k = \eta = \sqrt{\frac{2(f(x^0) - f(x^\star))}{LT\alpha}}$$

with

$$T \geq \frac{L\beta^2(f(x^0) - f(x^\star))}{\alpha}$$

(number of iterations), we have iteration complexity

$$\mathcal{O}\left(\sqrt{\frac{(\omega_W + 1)(\omega_M + 1)}{Tn}}\right),$$

which will be essentially same as doing no compression on master and using $\mathcal{C}_W \circ \mathcal{C}_M$ or $\mathcal{C}_W \circ \mathcal{C}_M$ on the workers' side. Our rate generalizes to the rate of Ghadimi & Lan (2013) without compression and dependency on the compression operator is better comparing to the linear one in Jiang & Agrawal (2018)[7]. Moreover, our rate enjoys linear speed-up in the number of workers $n$, the same as Ghadimi & Lan (2013). In addition, if one introduces mini-batching on each worker of size $b$ and assuming each worker has access to the whole data, then $\sigma^2 \to \sigma^2/b$ and $\zeta^2 \to 0$, which implies

$$\mathcal{O}\left(\sqrt{\frac{(\omega_W + 1)(\omega_M + 1)}{Tn}}\right) \to \mathcal{O}\left(\sqrt{\frac{(\omega_W + 1)(\omega_M + 1)}{Tbn}}\right),$$

and hence one can also obtain linear speed-up in terms of mini-batch size, which matches with Jiang & Agrawal (2018).

### D.7 SGD WITH BIDIRECTIONAL COMPRESSION: FOUR MODELS

It is possible to consider several different regimes for our distributed optimization/training setup, depending on factors such as:

- The relative speed of communication (per bit) from workers to the master and from the master to the workers,
- The intelligence of the master, i.e., its ability or the lack thereof of the master to perform aggregation of real numbers (e.g., a switch can only perform integer aggregation),
- Variability of various resources (speed, memory, etc) among the workers.

For simplicity, we will consider four situations/regimes only, summarized in Table 2.

**Direct consequences of Theorem 6**: Notice that (5) posits a $\mathcal{O}(1/T)$ convergence of the gradient norm to the value $\frac{\alpha L\eta}{2-\beta L\eta}$, which depends linearly on $\alpha$. In view of (4), the more compression we perform, the larger this value. More interestingly, assume now that the same compression operator is used at each worker: $\mathcal{C}_W = \mathcal{C}_{W_i}$. Let $\mathcal{C}_W \in \mathbb{B}(\omega_W)$ and $\mathcal{C}_M \in \mathbb{B}(\omega_M)$ be the compression on master side. Then, $T(\omega_M, \omega_W) := 2L(f(x^0) - f(x^\star))\varepsilon^{-2}(\alpha + \varepsilon\beta)$ is its iteration complexity. In the special case of equal data on all nodes, i.e., $\zeta = 0$, we get $\alpha = (\omega_M+1)(\omega_W+1)\sigma^2/n$ and $\beta = (\omega_M + 1)(1 + \omega_W/n)$. If no compression is used, then $\omega_W = \omega_M = 0$ and $\alpha + \varepsilon\beta = \sigma^2/n + \varepsilon$.

---

[7]Jiang & Agrawal (2018) allows compression on the worker side only.

So, the *relative slowdown* of Algorithm 1 used *with* compression compared to Algorithm 1 used *without* compression is given by

$$\frac{T(\omega_M,\omega_W)}{T(0,0)} = \frac{\big((\omega_W+1)\sigma^2/n + (1 + \omega_W/n)\varepsilon\big)}{\sigma^2/n + \varepsilon}(\omega_M+1) \in (\omega_M+1, (\omega_M+1)(\omega_W+1)]. \quad (17)$$

The upper bound is achieved for $n = 1$ (or for any $n$ and $\varepsilon \to 0$), and the lower bound is achieved in the limit as $n \to \infty$. So, *the slowdown caused by compression on worker side decreases with $n$.* More importantly, *the savings in communication due to compression can outweigh the iteration slowdown, which leads to an overall speedup!*

### D.7.1 MODEL 1

First, we start with the comparison, where we assume that transmitting one bit from worker to node takes the same amount of time as from master to worker.

| Compression $\mathcal{C} \in \mathbb{B}(\omega)$ | No. iterations $T(\omega) = \mathcal{O}((\omega+1)^{1+\theta})$ | Bits per iteration $W_i \mapsto M + M \mapsto W_i$ | Speedup $\frac{T(0)B(0)}{T(\omega)B(\omega)}$ |
|---|---|---|---|
| None | 1 | $2 \cdot 32d$ | 1 |
| $\mathcal{C}_{\mathrm{nat}}$ | $\left(\frac{9}{8}\right)^{1+\theta}$ | $2 \cdot 9d$ | 2.81×–3.16× |
| $S^q$ | $\left(\frac{d}{q}\right)^{1+\theta}$ | $2 \cdot (33 + \log_2 d)q$ | 0.06×–0.60× |
| $S^q \circ \mathcal{C}_{\mathrm{nat}}$ | $\left(\frac{9d}{8q}\right)^{1+\theta}$ | $2 \cdot (10 + \log_2 d)q$ | 0.09×–0.98× |
| $\mathcal{D}_{\mathrm{sta}}^{p,2^{s-1}}$ | $\left(1 + \sqrt{d}2^{1-s}\kappa\right)^{1+\theta}$ | $2 \cdot (32 + d(s+2))$ | 1.67×–1.78× |
| $\mathcal{D}_{\mathrm{nat}}^{p,s}$ | $\left(\frac{81}{64} + \frac{9}{8}\sqrt{d}2^{1-s}\kappa\right)^{1+\theta}$ | $2 \cdot (8 + d(\log_2 s + 2))$ | 3.19×–4.10× |

**Table 3:** Our compression techniques can speed up the overall runtime (number of iterations $T(\omega)$ times the bits sent per iteration) of distributed SGD. We assume *binary*32 floating point representation, bi-directional compression using $\mathcal{C}$, and the same speed of communication from worker to master ($W_i \mapsto M$) and back ($M \mapsto W_i$). The relative number of iterations (communications) sufficient to guarantee $\varepsilon$ optimality is $T'(\omega) := (\omega+1)^\theta$, where $\theta \in (1, 2]$ (see Theorem 6). Note that big $n$ regime leads to better iteration bound $T(\omega)$ since for big $n$ we have $\theta \approx 1$, while for small $n$ we have $\theta \approx 2$. For dithering, $\kappa = \min\{1, \sqrt{d}2^{1-s}\}$. The 2.81× speedup for $\mathcal{C}_{\mathrm{nat}}$ is obtained for $\theta = 1$, and the 3.16× speedup for $\theta = 0$. The speedup figures were calculated for $d = 10^6$, $p = 2$ (dithering), optimal choice of $s$ (dithering), and $q = 0.1d$ (sparsification).

### D.7.2 MODEL 2

For the second model, we assume that the master communicates much faster than workers thus communication from workers is the bottleneck and we don't need to compress updates after aggregation, thus $\mathcal{C}_M$ is identity operator with $\omega_M = 0$. This is the case we mention in the main paper. For completeness, we provide the same table here.

### D.7.3 MODEL 3

Similarly to previous sections, we also do the comparison for methods that might be used for In-Network Aggregation. Note that for INA, it is useful to do compression also from master back to workers as the master works just with integers, hence in order to be compatible with floats, it needs to use bigger integers format. Moreover, $\mathcal{C}_{\mathrm{nat}}$ compression guarantees free translation to floats. For the third model, we assume we have the same assumptions on communication as for Model 1. As a baseline, we take SGD with $\mathcal{C}_{\mathrm{nat}}$ as this is the most simple analyzable method, which supports INA.

### D.7.4 MODEL 4

Here, we do the same comparison as for Model 3. In contrast, for communication we use the same assumptions as for Model 2.

| Approach | $\mathcal{C}_{W_i}$ | No. iterations $T'(\omega_W) = \mathcal{O}((\omega_W + 1)^\theta)$ | Bits per 1 iter. $W_i \mapsto M$ | Speedup Factor |
|---|---|---|---|---|
| Baseline | identity | 1 | $32d$ | 1 |
| **New** | $\mathcal{C}_{\mathrm{nat}}$ | $(9/8)^\theta$ | $9d$ | $3.2\times$–$3.6\times$ |
| Sparsification | $\mathcal{S}^q$ | $(d/q)^\theta$ | $(33 + \log_2 d)q$ | $0.6\times$–$6.0\times$ |
| **New** | $\mathcal{C}_{\mathrm{nat}} \circ \mathcal{S}^q$ | $(9d/8q)^\theta$ | $(10 + \log_2 d)q$ | $1.0\times$–$10.7\times$ |
| Dithering | $\mathcal{D}_{\mathrm{sta}}^{p,2^{s-1}}$ | $(1 + \kappa d^{1/r} 2^{1-s})^\theta$ | $31 + d(2 + s)$ | $1.8\times$–$15.9\times$ |
| **New** | $\mathcal{D}_{\mathrm{nat}}^{p,s}$ | $(9/8 + \kappa d^{\frac{1}{r}} 2^{1-s})^\theta$ | $31 + d(2 + \log_2 s)$ | $4.1\times$–$16.0\times$ |

**Table 4:** The overall speedup of distributed SGD with compression on nodes via $\mathcal{C}_{W_i}$ over a Baseline variant without compression. Speed is measured by multiplying the # communication rounds (i.e., iterations $T(\omega_W)$) by the bits sent from worker to master ($W_i \mapsto M$) per 1 iteration. We neglect $M \mapsto W_i$ communication as in practice this is much faster. We assume *binary*32 representation. The relative # iterations sufficient to guarantee $\varepsilon$ optimality is $T'(\omega_W) := (\omega_W + 1)^\theta$, where $\theta \in (0, 1]$ (see Theorem 6). Note that in the big $n$ regime the iteration bound $T(\omega_W)$ is better due to $\theta \approx 0$ (however, this is not very practical as $n$ is usually small), while for small $n$ we have $\theta \approx 1$. For dithering, $r = \min\{p, 2\}$, $\kappa = \min\{1, \sqrt{d}2^{1-s}\}$. The lower bound for the Speedup Factor is obtained for $\theta = 1$, and the upper bound for $\theta = 0$. The Speedup Factor $\left(\frac{T(\omega_W) \cdot \# \text{Bits}}{T(0) \cdot 32d}\right)$ figures were calculated for $d = 10^6$, $q = 0.1d$, $p = 2$ and optimal choice of $s$ with respect to speedup.

| Approach | $\mathcal{C}$ | Slowdown (iters / baseline) | Bits per iter. $W_i \mapsto M + M \mapsto W_i$ | Speedup factor |
|---|---|---|---|---|
| **Baseline** | $\mathcal{C}_{\mathrm{nat}}$ | 1 | $2 \cdot 9d$ | 1 |
| **Sparsification** | $\mathcal{S}^q \circ \mathcal{C}_{\mathrm{nat}}$ | $(d/q)^{1+\theta}$ | $2 \cdot (10 + \log_2 d)q$ | $0.03\times$–$0.30\times$ |
| **Dithering** | $\mathcal{D}_{\mathrm{nat}}^{p,s}$ | $(9/8 + \kappa d^{\frac{1}{r}} 2^{1-s})^{1+\theta}$ | $2 \cdot (8 + d(2 + \log_2 s))$ | $1.14\times$–$1.30\times$ |

**Table 5:** Overall speedup (number of iterations $T$ times the bits sent per iteration ($W_i \mapsto M + M \mapsto W_i$) of distributed SGD. We assume *binary*32 floating point representation, bi-directional compression using the same compression $\mathcal{C}$. The relative number of iterations (communications) sufficient to guarantee $\varepsilon$ optimality is displayed in the third column, where $\theta \in (0, 1]$ (see Theorem 6). Note that big $n$ regime leads to smaller slowdown since for big $n$ we have $\theta \approx 0$, while for small $n$ we have $\theta \approx 1$. For dithering, we chose $p = 2$ and $\kappa = \min\{1, \sqrt{d}2^{1-s}\}$. The speedup factor figures were calculated for $d = 10^6$, $p = 2$ (dithering), optimal choice of $s$ (dithering), and $q = 0.1d$ (sparsification).

| Approach | $\mathcal{C}_{W_i}$ | $\mathcal{C}_M$ | Slowdown (iters / baseline) | $W_i \mapsto M$ commun. (bits / iteration) | Speedup factor |
|---|---|---|---|---|---|
| **Baseline** | $\mathcal{C}_{\mathrm{nat}}$ | $\mathcal{C}_{\mathrm{nat}}$ | 1 | $9d$ | 1 |
| **Sparsification** | $\mathcal{S}^q \circ \mathcal{C}_{\mathrm{nat}}$ | $\mathcal{C}_{\mathrm{nat}}$ | $(d/q)^\theta$ | $(10 + \log_2 d)q$ | $0.30\times$–$3.00\times$ |
| **Dithering** | $\mathcal{D}_{\mathrm{nat}}^{p,s}$ | $\mathcal{C}_{\mathrm{nat}}$ | $(9/8 + \kappa d^{\frac{1}{r}} 2^{1-s})^\theta$ | $(8 + d(2 + \log_2 s))$ | $1.3\times$–$4.5\times$ |

**Table 6:** Overall speedup (number of iterations $T$ times the bits sent per iteration ($W_i \mapsto M$) of distributed SGD. We assume *binary*32 floating point representation, bi-directional compression using $\mathcal{C}_{W_i}, \mathcal{C}_M$. The relative number of iterations (communications) sufficient to guarantee $\varepsilon$ optimality is displayed in the third column, where $\theta \in (0, 1]$ (see Theorem 6). Note that big $n$ regime leads to smaller slowdown since for big $n$ we have $\theta \approx 0$, while for small $n$ we have $\theta \approx 1$. For dithering, we chose $p = 2$ and $\kappa = \min\{1, \sqrt{d}2^{1-s}\}$. The speedup factor figures were calculated for $d = 10^6$, $p = 2$ (dithering), optimal choice of $s$ (dithering), and $q = 0.1d$ (sparsification).

### D.7.5 COMMUNICATION STRATEGIES USED IN TABLES 1, 3, 5, 6

**No Compression or $\mathcal{C}_{\mathrm{nat}}$.** Each worker has to communicate a (possibly dense) $d$ dimensional vector of scalars, each represented by 32 or 9 bits, respectively.

**Sparsification $\mathcal{S}^q$ with or without $\mathcal{C}_{\mathrm{nat}}$.** Each worker has to communicate a sparse vector of $q$ entries with full 32 or limited 9 bit precision. We assume that $q$ is small, hence one would prefer to transmit positions of non-zeros, which takes $q(\log_2(d) + 1)$ additional bits for each worker.

**Dithering ($\mathcal{D}_{\mathrm{sta}}^{p,s}$ or $\mathcal{D}_{\mathrm{nat}}^{p,s}$).** Each worker has to communicate $31(8 - \mathcal{D}_{\mathrm{nat}}^{p,s})$ bits (sign is always positive, so does not need to be communicated) for the norm, and $\log_2(s) + 1$ bits for every coordinate for level encoding (assuming uniform encoding) and 1 bit for the sign.

### D.8 SPARSIFICATION - FORMAL DEFINITION

Here we give a formal definition of the sparsification operator $\mathcal{S}^q$ used in Tables 1, 3,5,6.

**Definition 5** (Random sparsification). Let $1 \le q \le d$ be an integer, and let $\circ$ denote the Hadamard (element-wise) product. The random sparsification operator $\mathcal{S}^q : \mathbb{R}^d \to \mathbb{R}^d$ is defined as follows:

$$\mathcal{S}^q(x) = \frac{d}{q} \cdot \xi \circ x,$$

where $\xi \in \mathbb{R}^d$ is a random vector chosen uniformly from the collection of all binary vectors $y \in \{0,1\}^d$ with exactly $q$ nonzero entries (i.e., $\|y\|_0 = q$).

The next result describes the variance of $\mathcal{S}^q$:

**Theorem 10.** $\mathcal{S}^q \in \mathbb{B}(d/q - 1)$.

Notice that in the special case $q = d$, $\mathcal{S}^q$ reduces to the identity operator (i.e., no compression is applied), and Theorem 10 yields a tight variance estimate: $d/d - 1 = 0$.

*Proof.* See e.g. Stich et al. (2018)(Lemma A.1). $\qquad\square$

Let us now compute the variance of the composition $\mathcal{C}_{\mathrm{nat}} \circ \mathcal{S}^q$. Since $\mathcal{C}_{\mathrm{nat}} \in \mathbb{B}(1/8)$ (Theorem 1) and $\mathcal{S}^q \in \mathbb{B}(d/q - 1)$ (Theorem 10), in view of the our composition result (Theorem 3) we have

$$\mathcal{C}_W = \mathcal{C}_{\mathrm{nat}} \circ \mathcal{S}^q \in \mathbb{B}(\omega_W), \qquad \text{where} \qquad \omega_W = \frac{1}{8}\left(\frac{d}{q} - 1\right) + \frac{1}{8} + \frac{d}{q} - 1 = \frac{9d}{8q} - 1. \quad (18)$$

## E LIMITATIONS AND EXTENSIONS

Quantization techniques can be divided into two categories: biased (Alistarh et al., 2018; Stich et al., 2018) and unbiased (Alistarh et al., 2017; Wen et al., 2017; Wangni et al., 2018). While the focus of this paper was on unbiased quantizations, it is possible to combine our natural quantization mechanisms in conjunction with biased techniques, such as the TopK sparsifier proposed in Dryden et al. (2016); Aji & Heafield (2017) and recently analyzed in Alistarh et al. (2018); Stich et al. (2018), and still obtain convergence guarantees.

