# OpenReview forum: "Natural Compression  for Distributed Deep Learning"
_ICLR.cc/2021/Conference — Reject_

### Official Review · AnonReviewer1 · 2020-10-21
**Recommendation to Reject: Not enough contribution**

**Rating:** 5
**Confidence:** 5

**Review:**

This paper proposes a data compression method named natural compression. This method can be implemented very fast and could be added to other compression methods for additional compression. Then this is generalized to natural dithering for a more aggressive compression. In addition, it compares different compression methods for a bidirectional compressed SGD method.

Strength:

- This compression method is very fast and can be combined with other compression methods. This compression has a small relative variance. It also has advantages in hardware implementation.

Weakness:

- Except for the compression method, the optimization algorithms discussed in this paper is the standard parallel SGD. There are many advanced optimization algorithms in the literature, and it is more important to see its performance of this compression method on those advanced algorithms. Two examples are DoubleSqueeze (Parallel stochastic gradient descent with double-pass error-compensated compression) and DORE (A Double Residual Compression Algorithm for Efficient Distributed Learning), and both algorithms have bidirectional compression.
- The compression rate is large comparing to other data compression methods such as p-quantization in the paper (Distributed Learning with Compressed Gradient Differences). Also, the comparison in A.4 may be unfair for the standard dithering. For standard dithering, the variance is bounded, while the variance for natural compression is relatively bounded. Therefore, for large values, the variance of natural compression is larger than standard ones (the same applies to Theorem 2), while small values favor natural compression. In the analysis, the relative variance is applied, so the result does not favor standard ones. It would be more interesting to compare different compression methods in solving various types of machine learning problems with different algorithms.
- Table 1 may not reflect the speedup in real implementation. As mentioned above, the analysis is based on the relative variation, so it would be interesting to see the comparison in numerical experiments as well.
- Line 4 in the caption of Table 1. The second sentence is not completed.

---

> ### Author Response · Authors · 2020-11-22
> **This review makes many incorrect claims: we explain why**
>
> Thank you for your comments, feedback and time.
>
> Some of  the mentioned weaknesses are not justified as we argue below.
>
> - Yes, we analyze standard distributed SGD for nonconvex problems, but *with the use of bidirectional compression*. This is novel,  and is an important contribution on its own. However, we consider this a secondary contribution only - our paper stands on its own even without this contribution. Indeed, the main contribution are our new and highly effective compressors: natural compression and natural dithering, and the associated theory (e.g, provable exponential improvement on existing variants). We use our newly developed and analyzed SGD with bidirectional compression as a scientific testbed which enables us study and showcase the benefits of our proposed compression operators to other baseline compressors we are improving on. Similar improvements can be seen if we used a different optimizer instead. Note that we distinguish between an optimizer which takes a compressor as a hyper-parameter/input, and a compression operator. These can be mixed and matched in various ways. For example, all the methods mentioned by you can work with any (unbiased) compression operator and our key contribution—the highly efficient natural compression and dithering compressors—can be used within their algorithms as well, which expands their/our impact further. Moreover, the methods you mentioned are either later or concurrent contributions to our work, and hence it is not reasonable for us to compare to them.  As requested by the reviewer, we run extra experiments using TopK compression in conjunction with our natural compression to further support claims in our paper, please see our detailed [report](https://wandb.ai/-anonymous/natural_compression/reports/ICLR-2020--VmlldzozMzAyMzI?accessToken=azhpblkyj6xw2e2u7b25ozv2ealr63djb2gm2khloykscznid5ntlhfifcn0pq8h) that we are going to merge to the manuscript.
>
>
> - Both of the claims here are not correct. First of all, p-quantization from Mishchenko et al. 2019 is just a special case of natural dithering with p-norm and number of levels 1, thus one cannot say that the compression rate is large since this corresponds to the method proposed in our paper. Their main contribution is the development of the DIANA optimizer, which can be seen as a variance reduced version of distributed SGD for reducing variance coming from compression. Note that our compressors are perfectly compatible with a generalized version of DIANA developed later, which supports any unbiased compressor. We can reach any compression rate by composing our natural dithering with sparsification techniques (we offer a composition theorem), providing SOTA compression operators, see Figure 1. Your other claim is incorrect, too: the variance of standard dithering is not bounded and scales with the norm of compressed vector similarly to natural dithering. All the comparisons in A.4 or Theorem 2 are fair and they are independent of whether small or large values are communicated. The only case where natural dithering is worse than standard dithering is for the number of levels $s = \infty$ as natural dithering reduces to natural compression while standard dithering reduces to the identity operator. This is expected, explained by our theory, and is not really an interesting scenario as the number of communicated bits would be too large.
>
> - Yes Table 1, offering theoretical speedups, may not on its own automatically lead to speedup in practice. However, this is weak criticism as it applies equally to *all theoretical rates of all methods ever analyzed*.  So, this criticism should be discounted. Table 1 offers valuable theoretical insights in a meaningful communication cost scenario. Moreover, as we show, our theoretical insights are useful predictors of actual performance, and hence are valuable. Moreover,  theoretical guarantees are valuable on their own. Actual speedup in a practical implementation will depend on the system used, implementation, data set, choice of hyper parameters and so on - so, comparisons of this type have their own set of issues. In our paper we analyze 4 communication cost scenarios (Tables 1, 3, 5, 6) covering a range of situations. Practitioners can study these scenarios and consult the one most closely resembling the distributed compute architecture they use for training. Note that we provide guarantees for all 4 scenarios.
>
> - Thank you for spotting this typo, we will complete the sentence.
>
>
> We hope that we addressed all your concerns. Please let us know in case something is not clear. We request a substantial increase in score as we believe much if the criticism was factually invalid, irrelevant, or minor.

---

### Official Review · AnonReviewer2 · 2020-10-26
**My concerns are mainly about the the significance of the proposed method in practice and the comparison to the previous work**

**Rating:** 5
**Confidence:** 3

**Review:**

This paper proposes a novel unbiased bidirectional compressor for vanilla SGD to reduce the communication overhead. Both theoretical and empirical analysis are provided. In overall, the paper is technically sound.

My concerns are mainly about the the significance of the proposed method in practice and the comparison to the previous work:

1. Dist-EF-SGD [1] and SignSGD [2] both achieve nearly 32x bidirectional compression, while natural compression only achieves roughly 8x in the experiments. Although this paper focuses on unbiased compressors and Dist-EF-SGD and SignSGD are biased compressors, the gap in the compression ratio is hard to ignore, which makes it hard to claim that this paper achieves SOTA in practice.

2. For the CIFAR-10 experiments, all the models (the smallest is ResNet50) are heavily over-parameterized, which means that there is huge redundancy in the gradients and models themselves. As a result, it will be easy to converge to small training loss for any compressor. Furthermore, since the experiments lacks comparison to other works (the only baseline with compression is standard dithering), it's hard to justify the importance of the proposed compressor. It will be better if the authors could show results on simpler models such as resnet20.

3. The paper lacks comparison to other methods. I still highly recommend to compare to dist-EF-SGD [1], since it achieves SOTA performance in communication-efficient distributed SGD. For other compressors, I understand that the other unbiased compressors may not have bidirectional compression. However, SignSGD [2] achieves bidirectional 32x compression without error feedback (error feedback improves the convergence of SignSGD, but not neccessary in some cases). Although SignSGD is biased compressor, it satisfies the requirement "vanilla distributed SGD with bidirectional compression" mentioned in Section 4, which makes it a good baseline. However, SignSGD is not compared or cited in this paper.

4. Most of the experiments are relatively small. For the ImageNet experiments in the appendix, no comparison in training time is provided.


References:
[1] Zheng, Shuai, Ziyue Huang, and James Kwok. "Communication-efficient distributed blockwise momentum SGD with error-feedback." Advances in Neural Information Processing Systems. 2019.
[2] Bernstein, Jeremy, et al. "signSGD: Compressed Optimisation for Non-Convex Problems." International Conference on Machine Learning. 2018.

---

> ### Author Response · Authors · 2020-11-22
> **We believe many of the issues raised here are either very minor, or are not issues and merely reflect a misunderstanding of our work**
>
> Thank you for your valuable comments, feedback and time.
>
> Below please find our answers to your concerns.
>
> 1. It is true that our natural compression can reach compression ratio up to 8x. However, this is *not* the only method that we propose in this paper and we can go much beyond that; even more than 32x, e.g. using a combination of sparsification and natural dithering or compression. Moreover, the compression ratio is not the only thing that matters, which your comment completely ignores. Not that compression does not come for free: it increases variance of stochastic gradients (and more aggressive compression increases variance more), and this leads to an optimizer-dependent increase in the number of communication rounds. So, what matters is the combined effect of these two factors. Clearly, if only compression ratio mattered, the optimal choice would not be any of the methods you list, but compressing everything to 0. However, such a method would not work anymore. This is an extreme example which we use to make a point. For a fixed compression ratio (number of bits), the most important quantity is the extra variance term introduced by compression. We show that for our methods the variance is the smallest among the competing methods, thus leading to the best methods, see Figure 1. Note that SignSGD is a heuristic not even guaranteed to converge: there are counterexamples which show its possible divergence, e.g. https://arxiv.org/abs/1901.09847. Our approach is in a different category altogether. Re comparison to EF-SGD: this is just an optimizer that can work with any contractive compressor, and our compression methods (after scaling) are contractive and hence compatible with EF-SGD. So, EF-SGD is not in any way competing with our compressors. Instead, EF-SGD is an alternative to our distributed SGD with bidirectional compression; one that aims to solve the problem of working with *biased compressors*. Since all our compressors are unbiased, EF-SGD is a complementary method aiming to solve a different problem. It should also be noted that one of the most popular compressors used with EF-SGD--- the Top-K compressor---can be combined with our natural compressor, which further reduces the number of bits communicated by a factor of 8 with minimal increase of iterations (variance only grows by ~12%). As requested by the reviewer, we ran the extra experiments to further support our claims, please see the second comment.
>
> 2. We are sorry for the confusion here. The ResNet50 (>23 million parameters) experiments are on ImageNet. Please note that ResNets for CIFAR10 have far fewer parameters than ResNets for ImageNet, e.g ResNet110 for CIFAR10 has only 1.7 million parameter (ResNet paper: https://arxiv.org/pdf/1512.03385.pdf). Besides, we also have natural compression experiments with ResNet20 (0.27 million parameters) in appendix A.1, where we also have more computation-intensive models such as DenseNet. In addition, we perform experiments which  include a comparison of our “natural” approach for both dithering and sparsification techniques and they show significant benefits of using “natural” choice of the compressor, see Figures 8 and 16-20 (in the Appendix). Finally, a principled approach to deal with models that are not over-parameterized is called variance reduction (VR). The DIANA method of Mishchenko et al is distributed SGD method with compression able to iteratively reduce the variance introduced by gradient compression. This has the effect that it is possible to solve (convex and strongly convex) problems to optimality, even with compression. Our compressors are compatible with DIANA, which shows that they apply even in the non-overvarameterized regime if a suitable optimizer is chosen.
>
> 3. We would like to point out that the main contribution of our work is not bidirectional SGD, but our compression methods! These can be used with any algorithm, e.g. EF-SGD. Moreover, they can be used in conjunction with other popular methods such as Top-K sparsifier and still significantly reduce communicated bits with minimal effect on convergence. As mentioned before, we ran those experiments to further support these claims.
>
> 4. The main message that could be concluded from our experiments is the following. For every method, our “natural” version of any given compression (e.g. no compression -> natural compression or standard dithering -> natural dithering)  converges in (almost) the same number of iterations while reducing the number of communicated bits significantly. Moreover, the computational overhead of the “natural” version is minimal due to its compatibility with float numbers, thus for every problem where communication is a bottleneck, our methods bring the significant speedup in terms of time.
>
> We hope that we addressed all your concerns. Please let us know in case something is not clear!

---

> > ### Author Response · Authors · 2020-11-24
> > **Extra Experiments**
> >
> > As requested by the reviewer, we run additional experiments using TopK compression in conjunction with our natural compression to provide even more evidence in the support of the claims in our paper, please see our detailed [report](https://wandb.ai/-anonymous/natural_compression/reports/ICLR-2020--VmlldzozMzAyMzI?accessToken=azhpblkyj6xw2e2u7b25ozv2ealr63djb2gm2khloykscznid5ntlhfifcn0pq8h) that we are going to merge to the manuscript.

---

### Official Review · AnonReviewer4 · 2020-10-29
**A stochastic quantization scheme with log-scale quantization levels**

**Rating:** 5
**Confidence:** 4

**Review:**

The paper proposed a new compression scheme called natural dithering, which uses powers of 2 as quantization levels in gradient compression. The variance of the scheme is studied and the compression scheme is tested in neural network training experiments.

Pros:
The proposed compression scheme is easy to implement in practice and the experiment results show it can reduce training time in practice. The study on different aspects of natural dithering is quite comprehensive, including the variance, comparison with standard dithering, and the limit regime when the number of bits goes to infinity (where standard dithering is better).

Cons:
My main concern is the novelty of this paper. Although discussions in the paper are quite comprehensive, the proposed compression scheme is just changing the uniform quantization levels to powers of 2. The idea is quite trivial and standard especially given the current machine representation of floating-point numbers.

---

> ### Author Response · Authors · 2020-11-22
> **Your novelty concern is not justified - this review is superficial and not helpful**
>
> Thank you for your comments  and time. Your novelty concern is not justified as we argue below.
>
> You claim that our idea is trivial and and standard. This is false. Prior to our work, uniformly distributed quantization levels was the standard approach for gradient quantization. We noticed that replacing uniform by exponential quantization leads to an *exponential improvement in the variance* for any fixed level of compression, or equivalently, in *exponential reduction in the number of bits communicated* for any fixed level of variance. So, while the basic idea is simple in hindsight, which we also hope is due to our narrative optimized for intuitive and clear explanation of the ideas, it is not trivial, and should be measure by its effects! The effect is enormous (exponential improvement!) We find the idea elegant and beautiful rather than trivial.
>
> You claim our idea is standard: this is false. We are not aware of any prior work on gradient compression for distributed learning which would suggest and/or analyze our compression schemes. We note you did not provide any evidence for your claim, and we hence suggest this claim be retracted. Moreover, we find this review to be superficial, short, lacking any evidential support, and ignoring most of our contributions. Please read our paper again, we carefully list our contributions.
>
> Please note that besides natural dithering, our contribution also includes natural compression ($C_{\rm nat}$). $C_{\rm nat}$ introduces small variance ($1/8$) with $3.56 \times$ less communication for float 32, and is easy to implement. We also performed the first analysis of distributed SGD with bidirectional compression for nonconvex optimization. This is a secondary contribution in comparison to the impact our compressors are expected to have, but of significant value on its own nevertheless. We use this method as a method for testing various compressors and their combinations, in theory and practice. Our theoretical results suggest overall falser training in total communication complexity in several communication regimes. We also show that our compressors are combinable/composable with all unbiased compressors with bounded variance - and this substantially expands their use.
>
> We want to stress again that we do not agree that the *apparent* simplicity of our approach degrades its novelty. We exerted a substantial effort to make the ideas as easily digestible to the reader as possible, and this may have given the impression of simplicity. Do not be fooled by this; and we request that you view simplicity as an asset. Simplicity means that the methods will be more easily understood, adopted, and used. Simple solutions to complicated problems are preferable to complicated solutions, and are harder to obtain! Simplicity also implies compatibility with float representation of numbers, which makes our compression methods easy to implement. Our compressors guarantee SOTA variance bounds for any number of bits, thus having the smallest increase in the number of iterations; see Figure 1. Since the reviewer is calling our idea standard, we would like to ask him/her to give a list of works that proposed such an approach and/or show such strong theoretical/practical improvement compared to baselines.
>
> Please feel free to ask any further questions, we are happy to answer!

---

### Official Review · AnonReviewer3 · 2020-10-30
**Natural Compression for Distributed Deep Learning Review**

**Rating:** 6
**Confidence:** 4

**Review:**

This paper introduces a new, simple yet theoretically and practically effective compression technique: natural compression. Theoretically, the compression technique increases the second moment of the compressed vector by not more than the factor of 9/8. Empirically, the communications savings are substantial, leading to 3-4 times improvement in overall running time.

Overall, I think the paper is theoretically solid and empirically validated. I tend to acceptance.

Pros:
1.  Theoretically grounded and rigorous criterion for compressing the information. The authors also give the theoretical and geometric interpretation why the natural compression technique can reduce communication without sacrificing too much accuracy.
2. It theoretically demonstrates that the savings in communication due to compression can outweigh the iteration slowdown, which leads to an overall speedup. This is not common in existing compression work.
3. The simulation considers various distributed learning settings and several standard datasets. It also considers the compatibility of the proposed compression schemes to the current IEEE standard and current programmable network switches. This provides evidence on the practical impact of the proposed compression scheme.

Cons:
1.   The literature review seems outdated. Considering various compression techniques (quantization and sparsification) have been proposed during the last two years, more recent work needs to be discussed and compared, e.g.,

Qsparse-local-SGD: Distributed SGD with Quantization, Sparsification and Local Computations, NIPS 2019
Communication-Efficient Distributed Learning via Lazily Aggregated Quantized Gradients, NIPS 2019
Online Learned Continual Compression with Adaptive Quantization Modules, ICML 2020
Moniqua: Modulo Quantized Communication in Decentralized SGD, ICML 2020

2.  In the simulations, the baselines are lacking. Specifically, the proposed compression techniques only compare with non-compression schemes, which is not sufficient.

---

> ### Author Response · Authors · 2020-11-22
> **First issue is very minor and easily fixable, second issue raised is factually incorrect**
>
> Thank you for your valuable comments, feedback and time.
>
> Please see our response to your concerns below. In case something is still not clear, please let us know, we are happy to further address any of your concerns.
>
> 1. Thank you for listing these works, we’ll add a discussion comparing our approach to them. For Q-sparse-local-SGD, Communication-Efficient Distributed Learning via Lazily Aggregated Quantized Gradients, and Moniqua: Modulo Quantized Communication in Decentralized SGD, please note that our algorithmic focus in this paper is bidirectional SGD, which is distinct to these aforementioned methods. Moreover, these methods can work with any compression operator and our key contribution—the highly efficient natural compression and dithering compressors—can be used within their algorithms as well, which expands their/our impact further. Online Learned Continual Compression with Adaptive Quantization Modules is not addressing gradient compression, which makes it hard to compare to. Finally, our work predates all these works and hence was not influenced by them. The fact that our compressors are combinable with the methods proposed in these subsequent works is a testament to the utility of our proposed compression techniques.
>
> 2. This is actually a false claim. Contrary to this, we compare our proposed methods to several SOTA compression operators such as QSGD, Terngrad or sparsification techniques. We refer the reader to Figures 8 and 16-20 (in the Appendix). As requested by the reviewer, we ran experiments using TopK compression in conjunction with our natural compression to further support claims in our paper, please see our detailed [report](https://wandb.ai/-anonymous/natural_compression/reports/ICLR-2020--VmlldzozMzAyMzI?accessToken=azhpblkyj6xw2e2u7b25ozv2ealr63djb2gm2khloykscznid5ntlhfifcn0pq8h) that we are going to merge to the manuscript.
>
> Please note the first issue raised is minor if an issue at all. It is easily addressable. The second issue is factually wrong. We would kindly request that you re-evaluate our paper in the light of this response.

---

### Author Response · Authors · 2020-11-24
**Follow-up experiments**

We have run a set of experiments following the reviews' suggestion.

We show that $C_{nat}$ can be composed with Top-K and hence can reduce the communicated data volume without a drop in model performance.

We compare with Top-K, SignSGD, EF-SignSGD, and PowerSGD.

Details are available in the following [report](https://wandb.ai/-anonymous/natural_compression/reports/ICLR-2020--VmlldzozMzAyMzI?accessToken=azhpblkyj6xw2e2u7b25ozv2ealr63djb2gm2khloykscznid5ntlhfifcn0pq8h).

Please let us know if anything is unclear.

---

### Decision · Program_Chairs · 2021-01-07
**Final Decision**

**Decision:**

Reject

**Comment:**

Three reviewers provided negative reviews and the authors wrote detailed feedback. During the later discussion stages, the reviewers acknowledged that some concerns are alleviated (e.g. R1 raised score from 4 to 5), but two concerns still remain: i) the novelty is less clear to the reviewers; ii) the advantage over existing approaches is not strong enough. I personally think the first concern is somewhat subjective; for the second concern, the authors indeed added a few experiments on "comparison to Top-K, SignSGD, EF-SignSGD, and PowerSGD", but R1 still maintained that the comparison is "not well demonstrated", which I guess is because R1 views the two works mentioned in R1's review as "concurrent or later" while R1 pointed out in the discussion that those two works appear before Oct 2019.
R2 pointed out that the paper could have shown the advantage in one of two aspects (see below) but did not.
See the discussions of the three reviewers below.

R1
"The advantage of the proposed compression method over existing approaches is not well demonstrated."
R2:
"To justify the significance of the contribution, I think the paper should show at least 1 of the followings:
When approaching a very high compression ratio, the natural compressor has significantly better testing accuracy compared to the previous work, i.e., the proposed work could push the compression ratio higher than the previous work with nearly no accuracy loss. The authors could use the new compressor alone, or combine it with some other compressors such as top-k, as the authors mentioned in the answers. (However, according to the extra experiments, the testing accuracy is slightly better than top-k, and slightly worse than power-k. None of them makes a significant difference.)
The natural compressors can achieve similar accuracy compared to the previous work with the same compression ratio, but with much less computation overhead. (I think this one may work for this paper, as the authors mentioned in their answers. However, to justify such a contribution, we will need to check the training time compared to EF-SGD with top-k and power-k, which seems not included in the extra experiments.)"
R4:
"I stand with my point that the novelty is limited."

Overall, I think the paper might have presented an interesting idea, but unfortunately can not be accepted in the current form.